# UniRestore3D: A Scalable Framework for General Shape Restoration

**Yuang Wang**[1]  **Yujian Zhang**[1]  **Sida Peng**[1]  **Xingyi He**[1]  **Haoyu Guo**[1]
**Yujun Shen**[2]  **Hujun Bao**[1]  **Xiaowei Zhou**[1*]
[1]State Key Lab of CAD&CG, Zhejiang University    [2]Ant Group
{wangyuang,xwzhou}@zju.edu.cn

## Abstract

Shape restoration aims to recover intact 3D shapes from defective ones, such as those that are incomplete, noisy, and low-resolution. Previous works have achieved impressive results in shape restoration subtasks thanks to advanced generative models. While effective for specific shape defects, they are less applicable in real-world scenarios involving multiple defect types simultaneously. Additionally, training on limited subsets of defective shapes hinders knowledge transfer across restoration types and thus affects generalization. In this paper, we address the task of general shape restoration, which restores shapes with various types of defects through a unified model, thereby naturally improving the applicability and scalability. Our approach first standardizes the data representation across different restoration subtasks using high-resolution TSDF grids and constructs a large-scale dataset with diverse types of shape defects. Next, we design an efficient and noise-robust hierarchical shape generation model that enables effective defective shape understanding and intact shape generation. Moreover, we propose a scalable training strategy for efficient model training. The capabilities of our proposed method are demonstrated across multiple shape restoration subtasks and validated on various datasets, including Objaverse, ShapeNet, GSO, and ABO.

## 1 Introduction

Restoring complete and intact shapes from defective ones is essential for applications in virtual reality, robotics, and content generation. Defective shapes can arise from various sources, including depth sensor noise (Tölgyessy et al., 2021), ill-posed 3D reconstruction (Wu et al., 2023), intrinsic defects in geometric representations (Feng & Crane, 2024), and so on. Consequently, shape restoration encompasses various repair goals, such as completion, super-resolution, and denoising, as illustrated in Fig. 1. These diverse tasks require model capabilities such as semantic understanding of highly incomplete and noisy shapes without additional inputs, robustness to varying degrees of incompleteness and noise, and the ability to produce diverse restoration results that balance quality and fidelity — making it challenging to design a unified model for general shape restoration.

To simplify the problem, many existing works focus on specific shape restoration goals other than addressing general shape restoration. By leveraging advanced deep learning techniques, such as regression models (Dai et al., 2017; 2020; Rao et al., 2022; Huang et al., 2023) and generative models (Mittal et al., 2022; Cheng et al., 2023; Warburg et al., 2023; Chu et al., 2024; Ju et al., 2024), these methods have achieved remarkable results on specific restoration goals. However, focusing on specific goals limits the models' capabilities in two key aspects. First, the defective shapes often contain various artifacts simultaneously, such as incompleteness and noise from ill-posed 3D reconstruction. Models that handle only certain artifacts have limited performance and applicability in these scenarios. Second, models aimed at specific goals are trained on limited types of data, and their learned object priors are not shared across tasks, resulting in less training data diversity compared to general shape restoration.

In this paper, we aim to address the task of general shape restoration, which targets the repair of various forms of defective geometries with a unified model. Compared with previous works on

---

*Corresponding author.

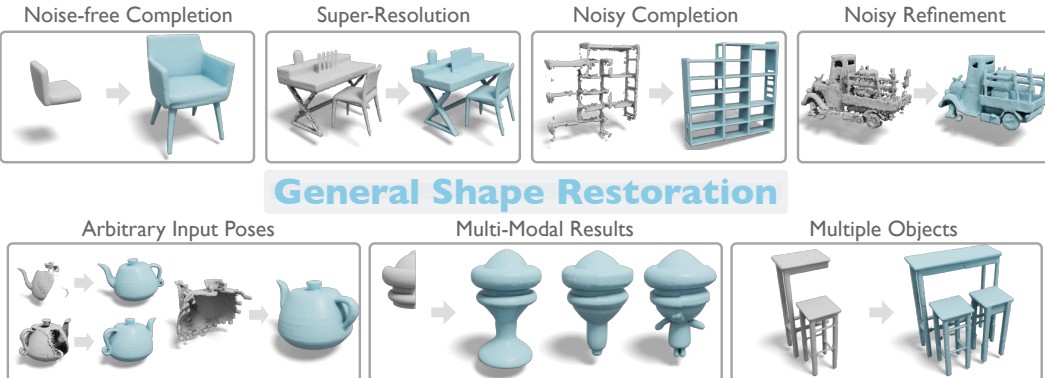

Figure 1: We propose UniRestore3D for general shape restoration. It can restore various types of defective 3D shapes under arbitrary poses, generate multiple possible restoration results, and support complex multi-object scenes.

specific restoration subtasks, the general shape restoration setting has two main advantages. First, it is more user-friendly. Users can apply a single model to repair various forms of defective geometries, which is significantly more convenient than first classifying the shape defect type and then using a mixture of corresponding restoration models. Second, this task enables us to exploit more data. By training on larger and more diverse datasets, the model can learn a more generalized shape prior, which is beneficial for handling in-the-wild defective shapes. As shown in Sec. 5.4, our model has better performance than models trained on specific subtasks.

To handle the general shape restoration task, we propose a novel, scalable framework that consists of three key components: a unified shape restoration dataset, an efficient and noise-robust shape restoration model, and a scalable training strategy. Regarding training data, as illustrated in Fig. 2, we construct synthetic defective-intact shape pairs with tailored construction methods for different subtasks, which realistically simulate real-world impairment scenarios. With the availability of large-scale datasets, model design and training strategies still face several technical challenges. First, restoring an intact shape solely from a defective one relies on the model's semantic understanding of the defective shape and its ability to efficiently generate high-quality shape. We design a hierarchical latent diffusion model (H-LDM) that achieves multi-scale encoding of defective shapes and efficient generation of intact ones, as shown in Fig. 3. Besides, two technical challenges remain. First, robustness to irregular noise in defective shapes; second, to ensure high fidelity to the existing input shape details, it is necessary to use high-resolution defective shapes as model inputs. This necessitates pre-compression of the defective shapes before training the H-LDM; otherwise, reading and encoding high-resolution shapes would unacceptably slow down training. Our key insight to solve these challenges is to construct a unified representation for both defective and intact shapes. Specifically, we first learn representations of intact shapes on large-scale data. Then, we train an defective shape encoder whose encodings are explicitly aligned to their corresponding intact ones. In this way, we achieve pre-compression of defective shapes, while enhancing noise robustness, as the feature alignment step enforces defective shape denoising.

The effectiveness of our method is validated on datasets including Objaverse (Deitke et al., 2023b;a), ShapeNet (Chang et al., 2015), ABO (Collins et al., 2022), GSO (Downs et al., 2022) and ScanNet (Rao et al., 2022). Our approach achieves SOTA results in tasks including noise-free shape completion, noisy shape refinement and completion. Related ablation studies have confirmed the effectiveness of key modules in our model.

# 2 RELATED WORK

## 2.1 3D SHAPE RESTORATION

Shape restoration encompasses various subtasks, each of which has been extensively studied in previous research. As illustrated in Fig. 2, the shape to be restored may exhibit local or extensive

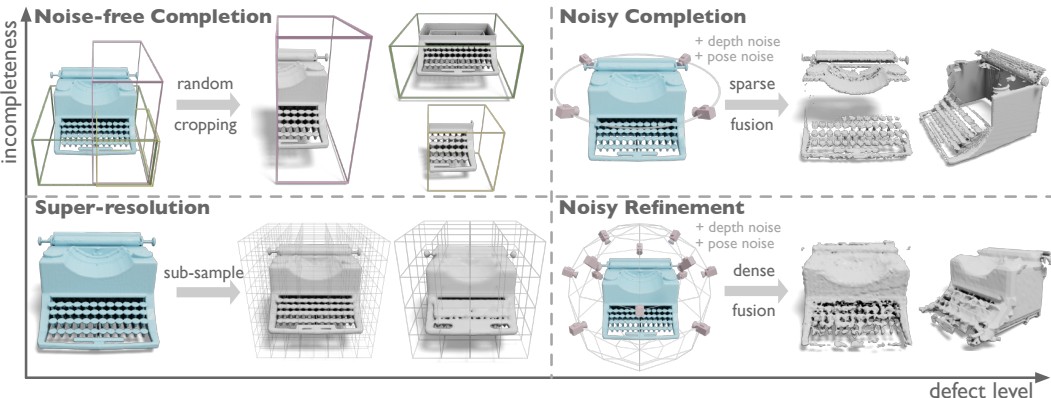

Figure 2: **Dataset curation and examples of shape restoration subtasks.** We create (defective, intact) shape pairs by adopting different impairment strategies for different subtasks.

missing regions, and the known areas may be noisy or noise-free. Most existing approaches focus on shape restoration under specific conditions, lacking general applicability. For example, surface reconstruction methods (Kazhdan & Hoppe, 2013; Peng et al., 2020; Huang et al., 2023) aim to restore complete geometry from noisy but fairly complete point clouds. Some works (Wu et al., 2018; Cui et al., 2023) target the restoration of complete point clouds from noisy ones with extensive incompleteness. Another line of works (Mittal et al., 2022; Li et al., 2023; Cui et al., 2024) train shape generative models to complete noise-free incomplete shapes. Among these, diffusion model based approaches rely on SDEdit (Meng et al., 2022) or blended diffusion (Lugmayr et al., 2022; Avrahami et al., 2022; 2023) mechanisms to complete the missing regions, which presents certain limitations in their use. Some works (Dai et al., 2017; Rao et al., 2022; Chu et al., 2024; Liu et al., 2024a; Galvis et al., 2024) aim to restore complete, noise-free TSDF grids from noisy and defective ones. However, they are trained on limited defect scenarios, which hinders their generality. Feng & Crane (2024) proposes a signed heat method for approximating the SDF of locally corrupted geometries, but does not leverage semantic information, thus limited to surface refinement and local restoration. Our goal is to build a conditional generative model for efficient, high-quality shape restoration that supports different types of impairment. Moreover, we emphasize the ability of accepting high-resolution defective shapes as inputs, which is necessary for high fidelity restoration.

## 2.2 3D SHAPE PRIOR

Shape restoration can be regarded as a form of shape prior for repairing defective geometries. Previous works have explored various types of shape priors and applied them to 3D reconstruction. By constraining 3D reconstruction to the latent or parameter space of shape prior models, these approaches help to avoid low-quality reconstruction results under ill-posed conditions (Zhu et al., 2018; Lin et al., 2019; Sucar et al., 2020; Liu et al., 2020; Yang et al., 2021; Sun et al., 2024). Among these studies, some object-level approaches can reconstruct complete 3D models under sparse observations but are limited to specific categories (Yang et al., 2022; Sucar et al., 2020). Part-level shape priors (Rao et al., 2022; Bokhovkin & Dai, 2023; Sun et al., 2022) are category-agnostic but lack semantic understanding and thus can not handle large missing regions. 3D diffusion models (DMs) have also been used as shape priors for 3D reconstruction (Warburg et al., 2023; Yang et al., 2023); however, they are still limited to modeling specific categories or small missing regions. General shape restoration aims to provide a general shape prior that is not confined to specific categories. We achieve this by training a conditional shape generative model, which directly maps the defective shapes onto the manifold of plausible shapes.

## 2.3 3D SHAPE GENERATIVE MODELS

3D shape generative models can be categorized based on the primary 3D representation utilized by the network, e.g., multiple 2D planes, hierarchical 3D structures, no explicit 3D representations, and hybrid approaches. To enhance generative modeling efficiency, these methods focus on leveraging

the sparsity of 3D data. Methods based on multi-view images (Szymanowicz et al., 2023; Shi et al., 2024; Liu et al., 2024b) or tri-planes (Chan et al., 2022; Gupta et al., 2023; Shue et al., 2023) exploit the sparsity of 3D data by reducing the dimensionality of the core representation, which effectively represent individual assets. Still, they are less efficient when handling complex, large-scale scenes. Approaches that discard 3D inductive bias and use latent features (Zhang et al., 2023; 2024) struggle with scaling feature sets and controlling their spatial distribution, limiting their applicability to large scenes. Hierarchical generative models based on sparse voxel hierarchies (Zheng et al., 2023; Ren et al., 2024) explicitly leverage sparsity by generating in a coarse-to-fine manner, which amortizes the generation process across levels, supports dynamic control of geometric details, and scales more efficiently to large scenes. The hierarchical structure naturally introduces multi-level conditional signals, facilitating a nuanced understanding of conditional inputs. We adopt a hierarchical generative model as our foundation and train a conditional model for general shape restoration.

## 3   GENERAL SHAPE RESTORAION

General shape restoration aims to restore complete and clean shapes $\mathbf{x}$ from incomplete and noisy inputs $\mathbf{x_c}$, which follows the conditional distribution $P(\mathbf{x} \mid \mathbf{x_c}, \mathbf{c})$ with optional conditions $\mathbf{c}$ like text. Its generality is demonstrated by its extensive support for defective inputs, encompassing regular or irregular geometric deficiencies and noise from sources such as sparse viewpoints, sensor noise, and reconstruction flaws. The main challenges include understanding various incomplete and noisy inputs, identifying regions needing restoration, accommodating shapes under different poses, and preserving the original structure and semantics while generating high-quality restoration results. Based on the level of noise and incompleteness present in the input geometry, we roughly categorize the general shape restoration into four subtasks: noise-free completion, super-resolution, noisy completion, and noisy refinement, among which previous works (Rao et al., 2022; Chu et al., 2024; Li et al., 2023) only focus on solving single specific subtasks.

**Dataset creation.** We represent both defective and intact shapes using the TSDF grid for its versatility. To construct numerous (defective, intact) shape pairs for model training, we employ different approaches for subtasks as shown in Fig. 2:

(1) Noise-free completion: Randomly sample partial shapes with varying levels of incompleteness from complete shapes, akin to image inpainting.
(2) Super-resolution: Subsample complete shapes to lower resolutions to create defective shapes of varying difficulty.
(3) Noisy completion: Render depth maps from sparse viewpoints, apply TSDF fusion to reconstruct incomplete shapes, and introduce varying degrees of noise into depth maps and camera poses to simulate geometric noises and missing scenarios.
(4) Noisy refinement: Similar to noisy completion but fuse input shapes from more viewpoints, resulting in complete structures with noisy surface details.

We randomly perturb model poses to enhance diversity and avoid canonical pose modeling. Since super-resolution does not conflict with the other subtasks, we randomly use different resolutions in the other three subtasks to enhance data diversity and increase task difficulty. Applying these strategies to datasets including Objaverse, ShapeNet, ABO, and GSO yields a large-scale dataset with approximately 120k object models and 800k shape pairs for training and evaluation.

## 4   METHOD

We devise a conditional shape generative model to achieve general shape restoration, which restores intact shapes from defective ones. Our model consists of three modules: a hierarchical variational autoencoder (H-VAE) for intact shapes compression, a hierarchical noise-robust encoder for processing defective shapes, and a hierarchical latent diffusion model (H-LDM) for conditional generation. We adopt a hierarchical approach to shape encoding and generation, enabling a multi-level understanding of defective geometries and efficient shape restoration. Fig. 3 presents the inference pipeline of shape restoration. Specifically, we first encode the defective input into multi-level sparse feature grids utilizing the noise-robust encoder. These feature grids serve as conditional inputs at different levels of the H-LDM, which generates sparse latent grids of intact shapes that can be further decoded to sparse TSDF grids from low to high resolution. In the following sections, we will

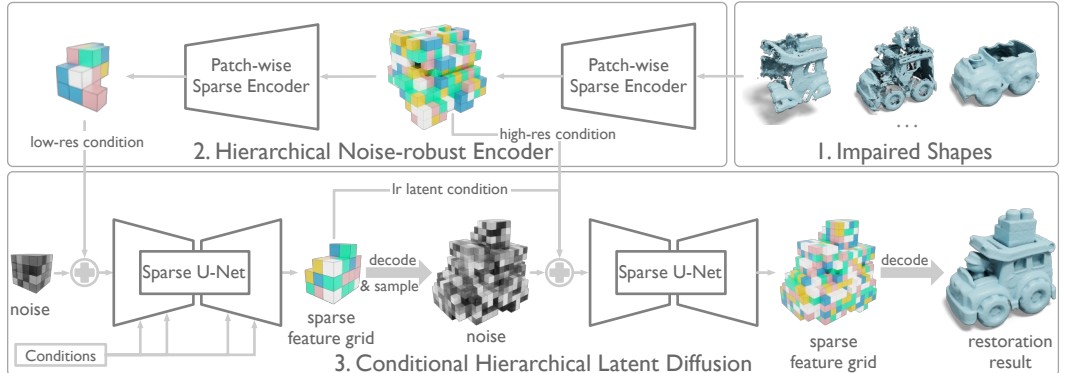

Figure 3: **Inference pipeline of the proposed shape restoration model.** We leverage a conditional hierarchical latent diffusion model (H-LDM) to restore complete and clean shapes from incomplete and noisy inputs. Defective input shapes are encoded into multi-level sparse features (two levels here) with a hierarchical noise-robust encoder, which acts as conditional signals of the H-LDM to generate multi-level sparse feature grids, and finally decoded to the restored shapes.

first introduce the hierarchical encoding of intact and defective shapes and the proposed scalable training strategy in Sec. 4.1 and then present the conditional H-LDM in Sec. 4.2.

## 4.1 HIERARCHICAL SHAPE ENCODING

The conditional H-LDM for shape restoration relies on compact encodings of intact and defective shapes. We propose a unified hierarchcial shape encoding pipeline for both the intact and defective shapes, which respectively acts as the generation targets and conditional inputs of the H-LDM. Here we first introduce the probabilistic modeling and architectural design, and then the scalable two-stage training strategy.

**Probabilistic modeling.** We use a cascaded VAE as in Razavi et al. (2019); Vahdat & Kautz (2020) to learn the multi-level latent representations jointly, instead of learning individual VAEs for shapes at different levels individually as in Ren et al. (2024). This design simplifies the training process and facilitates learning inter-level dependencies across multi-level latents. The H-VAE consists of a series of cascaded encoders $E = \{E^1, \ldots, E^L\}$ across levels and independent decoders $D = \{D^1, \ldots, D^L\}$ at each level, as shown in the top row of Fig. 4. Specifically, the H-VAE encoders learn the approximate posterior $q_\phi(\mathbf{z} \mid \mathbf{x}) = \prod_i q_\phi(\mathbf{z}^i \mid \mathbf{z}^{<i}, \mathbf{x})$, where each level's latent $\mathbf{z}^i$ is built upon the $\mathbf{z}^{<i}$ from previous levels. The H-VAE decoders learn the likelihood $p_\varphi(\mathbf{x} \mid \mathbf{z}) = \prod_i p_\varphi(\mathbf{x}^i \mid \mathbf{z}^{\geq i})$, where shape $\mathbf{x}^i$ at each level is decoded not only from the corresponding latent $\mathbf{z}^i$ but also based on the coarser levels $\mathbf{z}^{>i}$.

**Patch-wise encoding.** To enhance the generalization of shape encoders on both intact and impaired shapes, we employ a patch-wise encoding strategy. Previous methods (Mittal et al., 2022; Yan et al., 2022) utilize patch-wise encoding to ensure that encoders can support both complete and partial shapes without noise. We further extend the patch-wise encoder to simultaneously encode noisy and noise-free 3D shapes, reducing the risk of out-of-distribution (OOD) occurrences for noisy shape encodings. Specifically, we divide the 3D shapes into multiple non-overlapping patches and apply the aforementioned cascaded encoder to encode each patch into hierarchical sparse features. The encodings of all patches at each level are then concatenated to form the latent representations of the entire shape. The decoders at different levels are not restricted to patch-wise decoding, thereby avoiding inconsistencies at the boundaries between patches.

**Scalable training strategy.** Given a pre-trained H-VAE of intact shapes, we can efficiently train an unconditional H-LDM for shape generation. However, for a conditional H-LDM, we still need to train a defective shape encoder along with the H-LDM, which is extremely inefficient due to the need to load high-resolution defective shapes online. To mitigate this issue, we learn defective shape encodings in advance using a two-stage training strategy as shown in Fig. 4. Our key insight it to share an unified representation between intact and defective shapes by learning the hierarchical

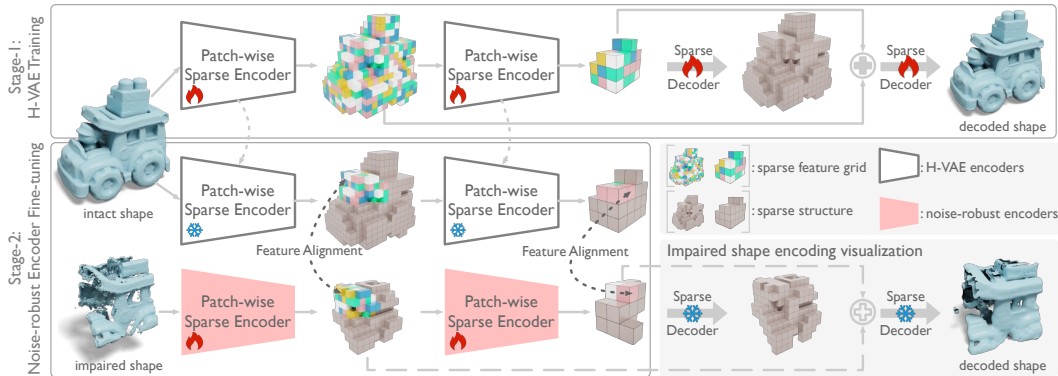

Figure 4: **Learning hierarchical shape encodings.** We learn hierarchical shape encodings of intact and defective shapes in two stages. These encodings are later used to train the conditional diffusion model for shape restoration. In stage-1 (top row), we train the hierarchical VAE (H-VAE) on intact shapes to learn hierarchical latent representations. In stage-2 (bottom left), we fine-tune a noise-robust encoder for encoding defective shapes through multi-level feature alignment between defective and intact encodings. The noise-robust encoder can robustly encode and pre-denoise the noisy defective shape as illustrated by the decoded shape (bottom right).

encodings of intact shapes first and then align the defective shape encodings to the intact ones. In this way, we can learn a defective shape encoder in advance and pre-compress shapes before LDM training. Moreover, the forced alignment leads to a noise-robust encoder as introduced below.

**Noise-robust encoder fine-tuning.** A naive approach to encoding impaired shapes is directly using the H-VAE encoders trained on intact shapes. However, given that the impaired shapes contain various types of noise that do not conform to a well-defined distribution, this approach is susceptible to the influence of OOD samples. To address this, we further perform a fine-tuning stage on the cascaded encoder of the pre-trained H-VAE and turn it into a separate noise-robust defective shape encoder $E_c$. Specifically, as shown in the bottom row of Fig. 4, given a pair of defective and intact shapes, we encode the defective shape with $E_c$ to multi-level encodings $\mathbf{z_c} = \{\mathbf{z_c}^1, \ldots, \mathbf{z_c}^L\}$, and encode the intact shape with the frozen pre-trained H-VAE encoder E to $\mathbf{z} = \{\mathbf{z}^1, \ldots, \mathbf{z}^L\}$. We train the noise-robust encoder $E_c$ by minimizing the discrepancy between these two sets of encodings. This fine-tuning stage not only enhances the robustness of the encoding of defective shapes but also pre-denoises the noisy inputs as shown in the bottom right of Fig. 4, thereby reducing the training difficulty of the subsequent conditional generative model.

## 4.2 HIERARCHICAL LATENT DIFFUSION MODEL

Given the hierarchical sparse encodings of the intact and defective shapes, we train a conditional hierarchical latent diffusion model (H-LDM) as the shape restoration model, which progressively generates 3D sparse structures and corresponding geometric attributes from low to high resolution. At each level, a sparse latent grid is generated and decoded to the concrete geometry, which serves as the sparse structure of the next level at higher resolution.

Specifically, given a intact shape $\mathbf{x}$ and a defective shape $\mathbf{x_c}$, we encode them separately to multi-level sparse encodings $\mathbf{z} = \{\mathbf{z}^1, \ldots, \mathbf{z}^L\}$ and $\mathbf{z_c} = \{\mathbf{z_c}^1, \ldots, \mathbf{z_c}^L\}$ as introduced in Sec. 4.1. We aim for the conditional H-LDM to learn the following probability distribution, which generates multi-level sparse latent grids from low ($i = L$) to high resolution ($i = 1$):

$$p_\theta(\mathbf{z} \mid \mathbf{z_c}, \mathbf{c}) = \prod_{i=L}^{1} p_\theta^i(\mathbf{z}^i \mid \mathbf{z}^{>i}, \mathbf{z_c}^i, \mathbf{c}), \qquad (1)$$

where $\mathbf{c}$ are optional conditions besides the defective shape $\mathbf{x_c}$, such as text. Notably, at each level, besides the conditional encoding $\mathbf{z_c}^i$ of the defective shape, we also rely on the previously generated coarser level latent grids $\mathbf{z}^{>i}$ as additional conditions to provide a global context for the generation of finer latent grids $\mathbf{z}^i$, which introduces more details.

We model each level's denoising process $p_\theta^i$ using a Sparse U-Net based denoiser, where its dependent sparse structure is obtained by decoding the sparse latent grid $\mathbf{z}^{i+1}$ from the previous level through the H-VAE decoder.

The H-LDM balances generation efficiency and quality thanks to the marrying of spatial sparsity and diffusion modeling on latent spaces. Moreover, it can explicitly reason over multi-level features of defective shapes through hierarchical conditioning for better restoration.

### 4.3 TRAINING AND IMPLEMENTATION DETAILS

Our pipeline consists of three trainable modules. For the H-VAE, we train all levels jointly using the standard ELBO objective (Kingma, 2013) at each level. For the H-LDM, we adopt a continuous-time diffusion model with $\mathbf{v}$-parameterization and use the simplified training objective (Ho et al., 2020) at each level. We fine-tune the noise-robust encoder with the following loss:

$$\mathcal{L}_{\text{align}} = \sum_{i=1}^{L} \left\| \mathbb{E}_{q_{\phi_c}(\mathbf{z}_{\mathbf{c}}^i | \mathbf{z}_{\mathbf{c}}^{\leq i}, \mathbf{x}_{\mathbf{c}})}[\mathbf{z}_{\mathbf{c}}^i] - \mathbb{E}_{q_\phi(\mathbf{z}^i | \mathbf{z}^{< i}, \mathbf{x})}[\mathbf{z}^i] \right\|_1, \tag{2}$$

which optimizes the noise-robust encoder to align the multi-level encodings of the defective shapes to the intact ones. We compute the loss only on sparse voxels shared by both defective and intact shapes. We employ sparse convolution (Tang et al., 2023) in all three modules for efficient processing of sparse voxel grids. However, at the coarsest level, the voxel grid becomes a dense one, where we use dense convolution for further processing. More details are provided in the appendix.

## 5 EXPERIMENTS

To validate the effectiveness of our proposed method, we conduct evaluations on several different shape restoration subtasks. *We use an unconditional model (i.e., with only shape condition) for shape restoration* without additional conditions like text to demonstrate the model's understanding capability of the impaired shapes. We first conduct experiments on our proposed general shape restoration task in Sec. 5.1, and then verify our model's effectiveness on existing benchmarks of shape restoration subtasks in Secs. 5.2 and 5.3. We train the model on a dataset that combines Objaverse and ShapeNet, comprising approximately 120k object models and 800k (defective, intact) shape pairs. Intact shapes are represented with $256^3$ TSDF grids, while impaired shapes, originally at various resolutions, are upsampled to $256^3$ and used as conditional inputs for the model. More details on data preprocessing and results postprocessing are provided in the appendix.

### 5.1 GENERAL SHAPE RESTORATION

The proposed task of general shape restoration contains different types of shape impairments. We evaluate the model separately on known categories and in-the-wild instances to quantify its capability. The known categories consist of 13 classes from ShapeNet with a substantial number of training samples. The in-the-wild categories refer to other unseen categories, which may have very few or no samples included in the training set. For known categories, we use the ShapeNet-13 (Liu et al., 2020) test set. For in-the-wild categories, we constructed the test set with GSO and ABO categories which are not included in ShapeNet-13.

**Evaluation metrics.** Following MSC (Wu et al., 2020), we evaluate the quality, diversity, and fidelity of the restoration results separately with the Minimum Matching Distance (MMD), Total Mutual Difference (TMD), and Average Matching Distance (AMD) metrics. For each impaired shape, we sample $k = 10$ samples in unit cubes and calculate these metrics. The reported metrics are multiplied by $10^3$. More details are provided in the appendix.

**Results.** We present both quantitative and qualitative results across multiple dataset subsets and restoration subtasks. Quantitative results in Tab. 1 highlight difficulty variations among different data and subtasks. Benefiting from the advantage of more training samples, the model performs better on known categories compared to in-the-wild ones. Among subtasks, noisy completion and noise-free completion demand stronger semantic understanding and generation capabilities on broader semantics from the model, resulting in generally worse metrics across several dataset sub-

Table 1: Quantitative results of general shape restoration.

| MMD ↓ / AMD ↓ / TMD ↑ | In-the-wild Categories | | Known Categories |
|---|---|---|---|
| | GSO | ABO | ShapeNet-13 |
| super-resolution | 0.174 / 0.213 / 0.192 | 1.177 / 1.358 / 0.255 | 0.174 / 0.315 / 0.251 |
| noise-free completion | 2.985 / 5.920 / 4.201 | 1.267 / 2.701 / 1.916 | 0.462 / 1.313 / 1.060 |
| noisy refinement | 0.344 / 0.466 / 0.395 | 1.195 / 1.513 / 0.419 | 0.204 / 0.304 / 0.240 |
| noisy completion | 0.922 / 1.467 / 1.046 | 1.201 / 1.875 / 0.873 | 0.319 / 0.648 / 0.506 |

Table 2: Quantitative results of noise-free shape completion on the 3DQD benchmark.

| Method | Resolution | Half | | | Octant | | |
|---|---|---|---|---|---|---|---|
| | | MMD ↓ | AMD ↓ | TMD ↑ | MMD ↓ | AMD ↓ | TMD ↑ |
| PoinTr | point cloud | 5.316 | N/A | N/A | 21.57 | N/A | N/A |
| SeedFormer | point cloud | 4.972 | N/A | N/A | 23.99 | N/A | N/A |
| AutoSDF | $64^3$ | 3.510 | 8.200 | 4.660 | 5.720 | 12.79 | 8.260 |
| 3DQD | $64^3$ | 2.933 | 6.302 | 4.780 | 4.690 | 10.93 | **9.600** |
| NeuSDFusion | sdf field | 2.290 | 5.900 | 4.760 | **3.030** | 9.590 | 8.320 |
| Ours | $256^3$ | 2.268 | 6.143 | **5.330** | 3.840 | 9.930 | 9.080 |
| Ours (more samples) | $256^3$ | **1.844** | **4.752** | 3.960 | 3.560 | **8.400** | 7.020 |

sets. Fig. 5 illustrates the model's performance on different subtasks, showing its ability to comprehend severely missing or noisy inputs and generate multi-modal outputs.

## 5.2 NOISE-FREE (MULTI-MODAL) SHAPE COMPLETION

We use the 3DQD (Li et al., 2023) benchmark to evaluate the multi-modal completion for noise-free partial shapes. 3DQD trains a category-conditional model on ShapeNet-13, using either half or octant samples as partial shapes, and evaluates MMD, TMD, and AMD metrics separately. To generate category-conditioned samples, we pre-train our model on Objaverse using text conditions based on captions provided by Cap3D (Luo et al., 2024) and fine-tune it on ShapeNet-13 using category names as the text conditions. Notably, we do not train our model solely on the noise-free completion subtask but jointly on all restoration subtasks, ensuring its generalizability, which cannot be effectively evaluated on the 3DQD benchmark. More details are provided in the appendix.

**Results.** Our method achieves SoTA completion quality (MMD) on the half subset, significantly outperforming previous baselines based on various 3D representations, as shown in Tab. 2. It also delivers better overall results (AMD) on the octant subset. In our vanilla training data, noise-free completion accounts for only a small fraction of the samples (approximately 1/8 of the entire dataset). As shown in *Ours (more samples)*, by introducing more noise-free completion samples, we can further improve the model's performance. As illustrated in Fig. 6, our method provides restoration that better preserve the given partial input and generate outputs with finer details. Conditional generative models are known to trade diversity for quality (Sadat et al., 2024; Ho & Salimans, 2022); our model similarly exhibits slightly lower diversity than baselines. However, TMD measures diversity without considering plausibility. Our method achieves more plausible restorations than baselines.

## 5.3 NOISY SHAPE COMPLETION

The PatchComplete (Rao et al., 2022) benchmark focuses on restoring noisy 3D scans of unknown categories. It includes synthetic and real-world subsets with severely incomplete, noisy partial shapes constructed by virtual rendering and depth fusion on ShapeNet, as well as cropping from ScanNet scans. Restoration quality is assessed by IoU and Chamfer Distance. Similar to 3DQD, we fine-tune our pre-trained model on PatchComplete's training set. The base model is trained on the Objaverse subset of our dataset, with all novel categories presented in the test set filtered.

**Results.** As shown in Tab. 3, our method achieves SoTA results on novel categories from both synthetic and real-world data. Notably, inputs of this benchmark are very low-resolution ($32^3$) TSDF grids. As illustrated in Fig. 7, our model can handle such low-resolution noisy inputs, generating high-resolution geometry while maintaining consistency with the partial ones. Unlike baseline methods, we do not leverage the observability information contained in the original TSDF grid but rely solely on the geometric information near the surface. This leads to a more general and challenging problem, yet the proposed method still yields superior results compared to baselines.

Table 3: Quantitative results of noisy shape completion on ShapeNet and ScanNet objects of novel categories. We leave the full results of each novel category to the appendix.

| CD↓ / IoU↑ | 3D-EPN | Auto-SDF | PatchComplete | DiffComplete | SC-Diff | Ours |
|---|---|---|---|---|---|---|
| ShapeNet (avg. of 8 categories) | 5.58 / 59.4 | 5.86 / 45.2 | 4.27 / 65.4 | 4.10 / 67.5 | 4.08 / 68.3 | **3.90 / 70.6** |
| ScanNet (avg. of 6 categories) | 9.09 / 44.0 | 8.90 / 38.9 | 7.52 / 49.5 | 7.18 / 51.3 | 7.04 / 51.9 | **6.75 / 53.3** |

Table 4: Ablation study on joint subtasks learning.

| | ABO (in-the-wild categories) | | ShapeNet-13 (known categories) | |
|---|---|---|---|---|
| MMD↓ / AMD↓ / TMD↑ | noisy refinement | noisy completion | noisy refinement | noisy completion |
| noisy refinement only | 1.53 / 1.75 / 0.258 | 4.25 / 5.44 / 0.653 | 0.278 / 0.360 / 0.195 | 1.84 / 2.49 / 0.408 |
| noisy completion only | 1.49 / 1.82 / 0.372 | 1.95 / 2.68 / 0.693 | 0.265 / 0.385 / 0.219 | 0.640 / 0.965 / 0.391 |
| Ours (joint training) | **1.19 / 1.51 / 0.419** | **1.20 / 1.87 / 0.873** | **0.204 / 0.304 / 0.240** | **0.319 / 0.648 / 0.506** |

Table 5: Ablation study on conditional encoder pre-alignment fine-tuning.

| MMD↓ / AMD↓ / TMD↑ | noisy refinement | noisy completion |
|---|---|---|
| w/o pre-alignment | 0.308 / 0.400 / **0.299** | 0.401 / 0.577 / **0.435** |
| w/ pre-alignment | **0.299 / 0.383** / 0.296 | **0.376 / 0.534** / 0.407 |

## 5.4 ABLATION STUDIES

**Effectiveness of joint training on subtasks.** In this ablation study, we aim to validate the effectiveness of joint training on multiple shape restoration subtasks compared to training each subtask individually. We train two models separately on our dataset's noisy completion subset and the noisy refinement subset. Then, we test their performance on each task, comparing the results with those of the jointly trained model. As shown in Tab. 4, joint training on a dataset composed of various shape restoration scenarios can effectively improve performance across different cases. Joint training proves effective for both rare and common categories.

**Effectiveness of conditional encoding pre-alignment.** In this experiment, we aim to validate the effectiveness of our proposed conditional encoder training mechanism. We compare two setups: one using the base VAE encoder directly without fine-tuning as the conditional encoder and the other applying our proposed pre-alignment fine-tuning strategy to it. Training the conditional encoder from scratch is excluded, as loading the uncompressed high-resolution conditional shapes during training significantly slows down training, making it impractical. We compare these setups on a single ShapeNet category. As shown in Tab. 5 pre-alignment fine-tuning yields better results on noisy, defective shapes with varying levels of incompleteness, i.e., noisy refinement and completion.

## 6 CONCLUSION

In this work, we unify multiple shape restoration subtasks and simulate diverse scenarios — such as varying degrees of incompleteness and noise — to build a cohesive synthetic dataset for training a general-purpose shape restoration model. We employ a conditional hierarchical latent diffusion model for shape restoration, enabling multi-level understanding of the defective shapes and efficient generation of the intact ones. Additionally, we enhance the model's robustness to extreme OOD inputs through patch-wise and noise-robust encoding. Benefiting from the proposed scalable training strategy, we can pre-compress defective shapes, significantly improving the training efficiency of the generative model. We demonstrate our model's effectiveness across different shape restoration subtasks on multiple datasets.

**Limitation and future works.** Our conditional diffusion model achieves high-quality results but sacrifices diversity for quality, a common tradeoff in such models. Although effective in certain multi-object scenarios, our method is limited by the training data and cannot handle large-scale scenes. Exploring geometric restoration in large scenes through compositional or holistic approaches is a promising direction. Restoring shapes with significant incompleteness requires the understanding and generation capabilities of broad semantics. Enhancing our model by training on larger-scale datasets could improve its robustness. Additionally, representing objects with sparse TSDF grids remains insufficiently compact; hence, shape restoration based on primitive-based or CAD-related representations is a potential area for further improvement.

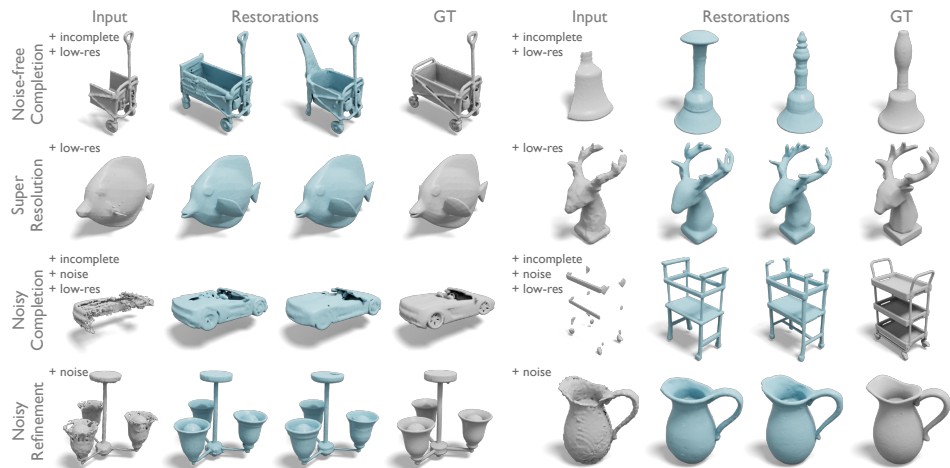

Figure 5: Qualitative results of shape restoration on our general shape restoration benchmark. UniRestore3D can handle various types of impairments with high quality.

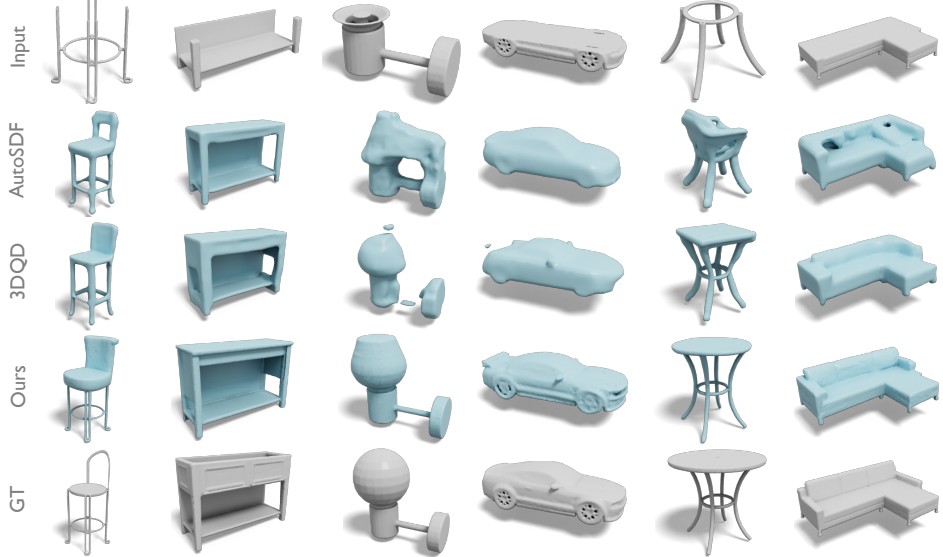

Figure 6: Qualitative results of noise-free shape completion on the 3DQD benchmark. UniRestore3D achieves better quality and fidelity compared to baselines.

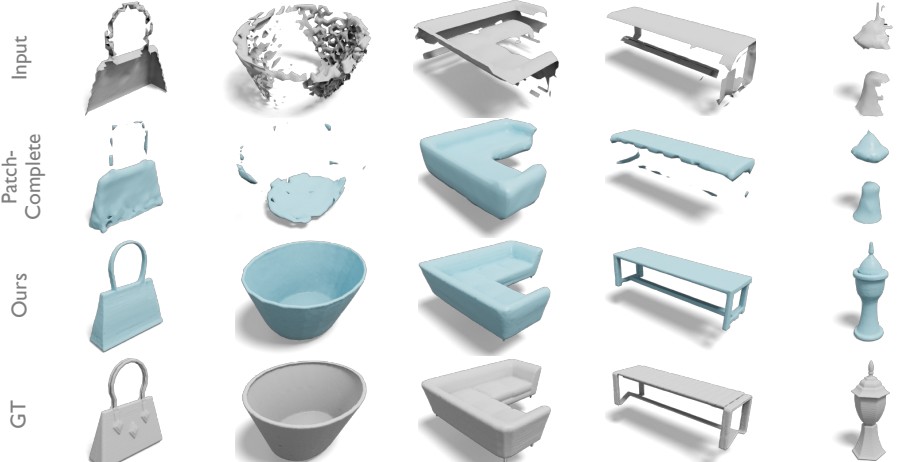

Figure 7: Qualitative results of noisy shape completion on the PatchComplete benchmark. UniRestore3D can restore low-resolution and noisy inputs effectively.

## ACKNOWLEDGEMENTS

This work was partially supported by the NSFC (No. U24B20154, No. 62322207, No. 62402427), Zhejiang Provincial Natural Science Foundation of China (No. LD25F030001), Ant Group Research Fund and Information Technology Center and State Key Lab of CAD&CG, Zhejiang University.

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

ETHICS STATEMENT

The proposed general shape restoration method and related datasets mostly do not involve human subjects or personal data; however, we recognize several potential ethical considerations. The constructed dataset is based on existing ones, including Objaverse (Deitke et al., 2023b;a), ShapeNet (Chang et al., 2015), ABO (Collins et al., 2022) and GSO (Downs et al., 2022), among which Objaverse includes 3d models of humans. The methods developed here are intended for general shape restoration, it is essential to consider the potential consequences if applied to sensitive areas, such as reconstructing protected or proprietary 3D models without authorization, 3D models involving human subjects, etc.

REPRODUCIBILITY STATEMENT

We provide detailed descriptions of each component of our method, including the dataset construction Sec. 3, the encoding methods and training strategy for defective and intact shapes Sec. 4.1, and the model and training procedure for the hierarchical latent diffusion model Sec. 4.2. The proposed method is built on publicly available codebases, including latent diffusion[1] and other open-source diffusion model implementations[2]. Main network architectures are built upon the publicly available TorchSparse codebase[3]. This information facilitates the reproduction of our method.

# A  APPENDIX

## A.1  MORE QUALITATIVE RESULTS

We present more qualitative results of our model on different subtasks of the proposed general shape restoration benchmark in Fig. 8.

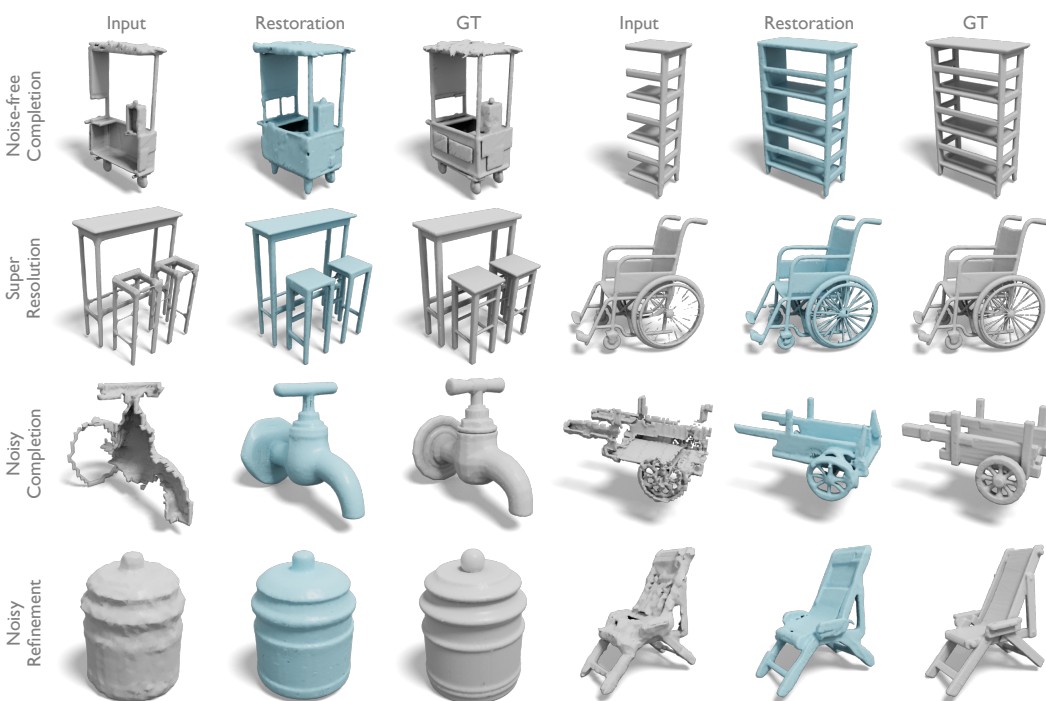

Figure 8: Qualitative results on our general shape restoration benchmark.

---

[1]https://github.com/CompVis/latent-diffusion
[2]https://github.com/lucidrains/denoising-diffusion-pytorch
[3]https://github.com/mit-han-lab/torchsparse

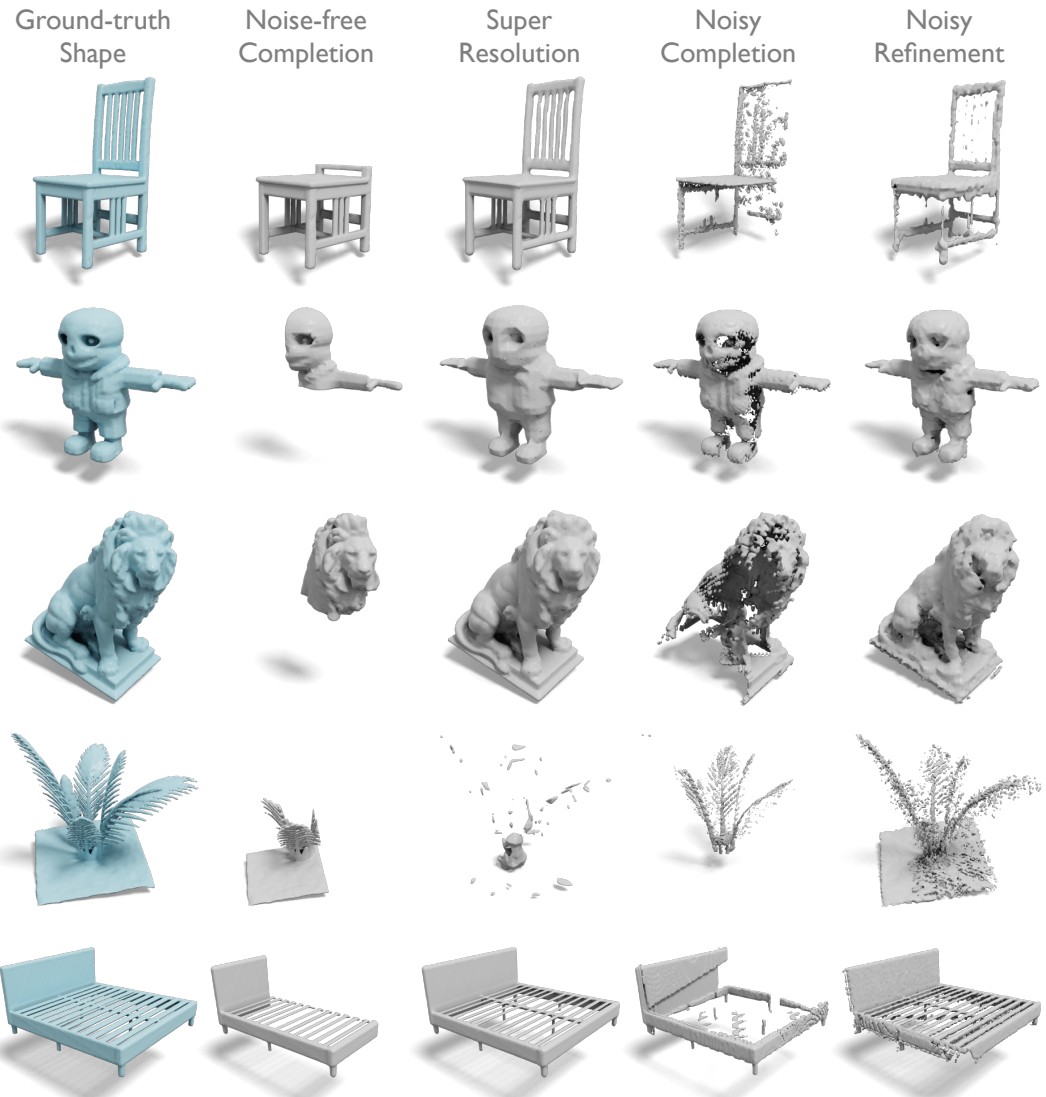

Figure 9: Data samples from the proposed general shape restoration dataset.

## A.2  NETWORK ARCHITECTURE

**Hierarchical VAE (H-VAE).** In the encoder part of the H-VAE, we adopt a cascaded structural design comprising multiple stages. First, for high-resolution inputs (e.g., $256^3$ or higher), we use a fast downsampling operation like $\mathrm{spatial2column}$ to reduce computational load and memory usage. This operation directly downsamples the input to the target resolution of the current stage encoder's bottleneck (e.g., from $256^3$ to $64^3$). This is equivalent to the use of a convolution layer for fast downsampling in Hoogeboom et al. (2023).

Then, we use $1 \times 1$ convolutions ($\mathrm{Conv}1 \times 1$) to encode the $\mathrm{spatial2column}$ results at each location into a feature space of dimension $C_1$. We then perform encoding using patch-wise residual blocks (ResBlocks) composed of sparse convolutions (without downsampling if $\mathrm{spatial2column}$ is used firstly), obtaining latent encodings of dimension $C_2$ for the current stage. Notably, if all values within a patch are truncated values, we skip the encoding and directly assign a globally-shared learnable feature. Proceeding to the next stage (lower resolution), we concatenate the sparse TSDF grid corresponding to this stage with the sparse latents from the previous stage as the input to this stage, and use $\mathrm{Conv}1 \times 1$ to encode it into a feature space of dimension $C_3$. Similar to the previous

Table 6: Training efficiency gains from precomputation using pre-trained noise-robust encoder.

| Model | w/ precomputation | | w/o precomputation | |
|---|---|---|---|---|
| | speed | time | speed | time |
| H-LDM (1st stage) | 4.37 it/s | 3.5 days | 0.22 it/s | 70 days |

stage, we use patch-wise ResBlocks composed of sparse convolutions (downsampling is performed here), obtaining latent encodings of dimension $C_4$ for the current stage. The specific structure of each ResBlock is the same as in the latent diffusion model (LDM) (Rombach et al., 2022).

In the decoder part, we use ResBlocks composed of sparse convolutions for decoding (no longer restricted to patch-wise). Each ResBlock's specific structure is the same as in LDM. We then use an MLP head to predict the TSDF values of the current stage and optionally a mask used to control sparse structure pruning, respectively. If the encoder corresponding to the current stage used the spatial2column operation for fast downsampling, we also additionally use the column2spatial operation during decoding for fast upsampling to obtain the relevant results at the target resolution.

**Hierarchical latent diffusion model (H-LDM).** In the H-LDM part, we employ an U-Net-based denoiser at each stage, where the U-Net is implemented using sparse convolution. The structure of the U-Net is consistent with that of the latent diffusion model (Rombach et al., 2022). The key difference is that the latents generated from the previous stage (which are at a lower resolution) are upsampled and used as additional conditional input for the current stage; this conditional input is modeled through concatenation. When using text as an additional conditional input — in experiments on the 3DQD benchmark — we inject the condition via cross-attention.

### A.3 TRAINING AND INFERENCE DETAILS

All our trainings conducts on 8 A100 GPUs. The experimental results in the paper are based on using a two-stage H-LDM; therefore, the model training is divided into four parts: training of the H-VAE, training of the noise-robust shape encoder, and the two-stage training of the H-LDM. Below are the training details of each sub-model:

**H-VAE.** The training of the H-VAE does not use data augmentation (since the VAE has strong generalization capabilities, and online pose augmentation for high-resolution TSDF grids is too time-consuming). We use 8 A100 GPUs to train the H-VAE for 200 epochs. The batch size is set to $8 \times 8 = 64$, and each iteration takes about 5 seconds. The training set (Objaverse subset) contains approximately 120K samples, and the training process takes about 14 days.

**Noise-Robust Encoder.** The training of the noise-robust encoder also does not use data augmentation. We use 8 A100 GPUs to train for 90 epochs. The batch size is $8 \times 8 = 64$, each iteration takes about 4.5 seconds, and the entire training process takes about 6 days.

**H-LDM (1st stage).** This part is trained with random object poses, we use 8 A100 GPUs for 400 epochs. The batch size is $32 \times 8 = 256$, with approximately 4.37 iterations per second, and GPU memory usage is about 27 GB. The training set contains approximately 800K samples, and the entire training process takes about 3.5 days. Our pre-trained noise-robust encoder enables precomputing compressed latents of defective shapes, significantly reducing I/O overhead for high-resolution shapes. This optimization accelerates training speed by approximately $20\times$, as demonstrated in Table 6.

**H-LDM (2nd stage).** Also trained with random object poses, we use 8 A100 GPUs for 200 epochs. The batch size is $32 \times 8 = 256$, with approximately 1.5 iterations per second, GPU memory usage is about 35 GB, and the entire training process takes about 5 days.

**Inference details.** We employ the DDIM sampler with 100 timesteps for diffusion sampling in all stages. For inference on an NVIDIA A100 GPU, the 1st stage denoiser runs approximately 10 timesteps per second, resulting in about 10 seconds per sampling and a VRAM usage of 8 GB. For the 2nd stage model, the sampling speed is around 6.5 timesteps per second, leading to an inference time of approximately 15 seconds per sampling and a VRAM consumption of 12 GB.

### A.4 EVALUATION PROTOCOL

**Evaluation metrics.** Following MSC (Wu et al., 2020), we evaluate the quality, diversity, and fidelity of the restoration results. For each impaired shape, we preprocess it and generate restoration results in unit cubes. Due to the intrinsic multi-modal distribution of shape restoration results, we sample $k = 10$ samples and calculate the following metrics. The Minimum Matching Distance (MMD) measures the quality of the restorations with the minimum Chamfer Distance (CD) between the $k$ samples and a single GT. The Total Mutual Difference (TMD) measures diversity using the average CD between each of the $k$ samples and the other $k - 1$ ones. The sum of $k$ average CDs is defined as TMD. TMD and MMD are strongly correlated; due to the inherent uncertainty in shape restoration, a model must generate diverse samples to achieve lower MMD. Ideally, fidelity should be assessed using the Unidirectional Hausdorff Distance (UHD), which measures the unidirectional distance from the given partial shape to the restoration result. However, since the entire shape may be impaired, leaving no intact region, computing UHD is not plausible, we approximate fidelity using the Average Matching Distance (AMD), which computes the average of the CD between the $k$ samples and the GT.

**Adapting pre-trained models to existing benchmarks.** Existing benchmarks only handle low-resolution shapes; e.g., PatchComplete (Rao et al., 2022) handles $32^3$ resolution TSDF grids, and 3DQD (Li et al., 2023) handles $64^3$ ones. Similar to image super-resolution, we upsample these low-resolution grids to $256^3$ as inputs. During evaluation, they are downsampled back to the required resolution.

### A.5 DETAILS OF GENERAL SHAPE RESTORATION DATASET

In this section, we provide a more detailed description of the construction method for our general shape restoration dataset. We present data samples from the dataset in Fig. 9. We use a 256-resolution TSDF grid to represent both the defective and intact shapes. While we consider defective shapes at different resolutions, similar to image super-resolution, they are all upsampled to 256-resolution voxel grids as inputs to the model. Low-resolution geometric representations inherently lead to geometric deficiencies. Beyond that, we consider three additional factors contributing to geometric degradation, combining them with different low-resolution representations to create more complex shape restoration scenarios. These scenarios correspond to noise-free completion, noisy completion, and noisy refinement. Below, we will describe the methods for constructing defective shapes for each case in detail.

**Intact TSDF grids computation.** When computing ground truth TSDF grids for intact shapes, we first use Manifold (Huang et al., 2018) to convert the original 3D models into watertight manifold surfaces. Next, we compute the distances based on point-mesh distances and determine the sign using fast winding numbers (Barill et al., 2018; Williams, 2022).

**TSDF grids upsampling.** We standardize various types of impaired shapes at different resolutions to a 256-resolution voxel grid as model inputs. To reduce artifacts caused by interpolation during the upsampling of low-resolution TSDF grids, we first extract the surface meshes and then recalculate the high-resolution TSDF grids. For the inside-outside determination, we use the pseudo-normal check (Bærentzen & Aanaes, 2005) instead of the generalized winding number (Jacobson et al., 2013), as we empirically found the pseudo-normal test leads to more plausible results in our case.

**Shape super-resolution.** Shape super-resolution aims to restore high-resolution shapes from ones represented in lower resolutions, which might exhibit detail loss, geometric artifacts, surface offset, etc. In the super-resolution subtask of our general shape restoration dataset, we construct geometries at multiple resolutions, including `{32, 64, 128, 256}`. We calculate TSDF grids at different resolutions using a method similar to that used for intact TSDF grids. Specifically, we first compute the watertight manifold surface based on the Manifold algorithm (Huang et al., 2018), with the parameter set for the highest resolution (256). This parameter corresponds to the depth of the octree constructed in intermediate steps, which influences the final geometric detail kept in the manifold surface. Based on the watertight manifold surface, we compute TSDF grids at different resolutions and then upsample them to 256-resolution voxel grids. We do not adjust the Manifold parameter for different resolutions, as this would create noticeable offsets between the resulting surfaces at different resolutions. While this may introduce more artifacts in low-resolution TSDF grids, we expect the shape restoration model to correct these artifacts.

Table 7: Quantitative results of noisy shape refinement across resolutions.

| Resolution | AMD $\downarrow$ | MMD $\downarrow$ | TMD $\uparrow$ |
|:---:|:---:|:---:|:---:|
| 32 | 0.518 | 0.347 | 0.489 |
| 64 | 0.457 | 0.357 | 0.358 |
| 128 | 0.463 | 0.353 | 0.373 |
| 256 | 0.211 | 0.161 | 0.203 |

**Noise-free shape completion.** Noise-free shape completion aims to recover a complete geometry from a partial shape with missing regions but no noise in the given areas. Similar to previous related works (Mittal et al., 2022; Cheng et al., 2023; Li et al., 2023), we randomly sample half or octant partial shapes from complete geometries to create the impaired shapes for restoration. We randomly sample complete TSDF grids at different resolutions and then randomly extract partial TSDF grids from them. These partial grids are upsampled to a resolution of 256 to increase diversity. We can extend the sampling of partial shapes to more random, irregular voxel grid sampling, thereby creating a wider range of data variations.

**Noisy shape completion.** Noisy shape completion aims to restore complete, noise-free geometry from incomplete geometries that exhibit noise and significant missing regions. Such geometries are common in 3D reconstruction, especially in cases of sparse viewpoints, insufficient network capacity, or significant errors in camera poses. To simulate these conditions, we employ depth fusion based on TSDF grids. We generate incomplete geometries by sampling sparse camera viewpoints and introducing random noise to depth maps and camera poses. Specifically, to simulate varying levels of sparse view observations, we apply several strategies: (1) using depth maps from a single view; (2) restricting camera viewpoints to cover a small, localized area, such as within a narrow cone; and (3) applying random cropping to the sampled sparse views to mimic random wide-baseline scenarios. On top of these viewpoint setups, we introduce random noise to both the camera poses and the rendered depth maps. For the depth maps, we simulate Kinect-like noise (Tölgyessy et al., 2021) with a probability of 0.5. For the camera poses, we randomly perturb both the rotation and translation components with a probability of 0.5.

**Noisy shape refinement.** Noisy shape refinement is similar to noisy shape completion but assumes the geometry to be repaired is relatively complete, with the primary task being the local optimization of surfaces affected by significant noise. This type of geometry frequently occurs in 3D reconstruction, for instance, due to camera pose errors, depth noise, or missing geometric details caused by algorithmic limitations. To simulate noisy shape refinement, we use the same camera pose and depth noise models as in noisy shape completion but ensure that the camera viewpoints fully cover the entire model.

**Impact of combining different shape degradation factors.** We categorize shape restoration into four subtasks. However, because different degradation factors are combined with randomly sampled resolutions to form defective shapes, there is no clear difficulty ranking among these subtasks. For example, in noisy shape refinement, even though more camera views are used in data generation than in noisy shape completion, the lower shape resolution can still lead to significant surface loss. In Tab. 7, we present the evaluation results of defective shapes at different resolutions for the noisy shape refinement subtask. It can be seen that, as the resolution increases, the model's AMD and MMD metrics generally improve, which means that higher resolutions result in less geometric loss and a simpler task.

## A.6 DETAILS OF 3DQD BENCHMARK

Similar to our training data, the 3DQD (Li et al., 2023) benchmark uses the TSDF grid as 3D representation for shapes. However, its data preprocessing method (calculating TSDF grids from possibly non-manifold meshes) differs from ours, and the resulting resolution of voxel grids is also different. These differences result in significant variations between the datasets, which can affect model evaluation. In the following, we will provide a detailed description of the data preprocessing and evaluation methods used in the 3DQD benchmark.

**Data preprocessing.** The 3DQD benchmark uses SDF grids provided by DIST (Liu et al., 2020) for both training and testing. The SDF grid is computed using the method proposed by Xu & Bar-

bič (2020). For a potentially non-manifold mesh, this method first calculates the unsigned distance function (UDF) and extracts an offset manifold surface, from which the SDF is then computed. This method generally produces a well-defined offset surface, but the surface corresponding to `level=0` often suffers from significant deficiencies. The 3DQD benchmark computes SDF grids with a resolution of 64, due to the low resolution, these deficiencies are further amplified. Consequently, the 3DQD benchmark adopts a strategy of computing relevant metrics based on the offset surface. Specifically, for a complete shape that is approximately normalized to fit within a unit cube (with some padding reserved), the 3DQD benchmark extracts surface meshes from both the GT SDF grid and the completion result at `level=0.04` and samples point clouds to compute relevant metrics.

Our model preprocesses defective shapes into TSDF grids with a resolution of 256 as model inputs. To avoid shape deficiency when extracting surface meshes at `level=0` for upsampling shapes from the 3DQD benchmark, we rebuild partial TSDF grids at resolution of 256, keeping the missing regions consistent with ones in 3DQD. For the model output, we first extract surface meshes at `level=0` and then recompute the SDF grid at lower resolution using the method proposed by Xu & Barbič (2020) to match 3DQD benchmark's required format.

**Training data.** The 3DQD benchmark evaluates a model's ability to complete shapes that are incomplete but free of noise. In contrast, our model aims for general shape restoration. Therefore, during training, we persist in using data from various restoration scenarios to maintain the model's versatility, and then evaluate the model on the 3DQD benchmark. Specifically, we preprocess shapes in the 3DQD benchmark's training set using the strategy shown in Fig. 2 to construct a general shape restoration data for model training. Moreover, due to the extreme incompleteness of shapes in the 3DQD benchmark, the object poses of the completion results tend to be highly diverse (e.g., when only chair legs are provided, the chair's pose can be ambiguous). Training the model under random poses often leads to significant differences between the predicted poses and the canonical pose of the ground truth. To address this, we train and evaluate our model in the canonical pose, consistent with other baselines.

### A.7 DETAILS OF PATCHCOMPLETE BENCHMARK

The PatchComplete benchmark (Rao et al., 2022) generates noisy and incomplete partial shapes for restoration using TSDF fusion. The corresponding ground truth (GT) TSDF grids are also obtained through TSDF fusion, but under dense viewpoints. Both the partial and GT TSDF grids are represented using a 32-resolution voxel grid. Due to its low resolution, the incomplete shape in PatchComplete contains limited geometric details. Below, we provide a detailed description of the data preprocessing and evaluation methods used for the PatchComplete benchmark.

**Data preprocessing.** Ideally, we could upsample the low-resolution TSDF grid from PatchComplete to the desired resolution for shape restoration. However, this straightforward approach can lead to inconsistencies in shapes' scale. Specifically, PatchComplete does not allocate sufficient padding during TSDF fusion, which may cause the object's geometric border to extend beyond the grid, resulting in an incomplete extracted mesh. Therefore, we cannot simply rescale the extracted mesh to fit entirely within the TSDF grid and then upsample it to a higher resolution. Instead, we adopt the same TSDF fusion method as PatchComplete, ensuring enough padding is reserved during fusion. After performing fusion within the 32-resolution TSDF grid, we upsample it as the input for our model.

Another issue in preprocessing the PatchComplete data (the ShapeNet subset) arises from its use of single-view depth for TSDF fusion. Directly extracting the mesh with marching cubes often leads to erroneous surfaces at the boundary between observed and unobserved regions, which corresponds to the camera's view frustum. We remove these erroneous surfaces before upsampling. In general, the TSDF grid obtained through TSDF fusion contains observability information, which shape completion methods with dense grids can directly use as inputs. These methods can retain the observed regions and only complete the non-observed areas. However, our model aims to achieve more general shape restoration. It uses a sparse voxel grid as input and only relies on information near the surface, without leveraging observability data, thus addressing a more challenging problem.

**Training data.** When training models for the PatchComplete benchmark, we fine-tune a pre-trained model using limited data available in PatchComplete. Since the benchmark emphasizes completion for novel categories, we exclude all instances related to the test set categories during pre-training on

Table 8: Quantitative results of noisy shape completion on ShapeNet objects of novel categories.

| CD↓ / IoU↑ | 3D-EPN | Auto-SDF | PatchComplete | DiffComplete | SC-Diff | Ours |
|---|---|---|---|---|---|---|
| Bag | 5.01 / 73.8 | 5.81 / 56.3 | 3.94 / 77.6 | 3.86 / 78.3 | 3.79 / 78.3 | **3.50** / **79.8** |
| Lamp | 8.07 / 47.2 | 6.57 / 39.1 | **4.68** / 56.4 | 4.80 / 57.9 | 4.74 / **60.0** | 5.35 / 59.7 |
| Bathtub | 4.21 / 57.9 | 5.17 / 41.0 | 3.78 / 66.3 | 3.52 / 68.9 | 3.67 / 65.9 | **3.39** / **72.0** |
| Bed | 5.84 / 58.4 | 6.01 / 44.6 | 4.49 / 66.8 | **4.16** / 67.1 | 4.40 / **67.1** | 4.46 / 66.5 |
| Basket | 7.90 / 54.0 | 6.70 / 39.8 | 5.15 / 61.0 | 4.94 / 65.5 | 4.89 / 68.5 | **3.70** / **74.2** |
| Printer | 5.15 / 73.6 | 7.52 / 49.9 | 4.63 / 77.6 | 4.40 / 76.8 | 4.36 / 76.8 | **4.18** / **80.5** |
| Laptop | 3.90 / 62.0 | 4.81 / 51.1 | 3.77 / 63.8 | 3.52 / 67.4 | 3.41 / 68.4 | **3.00** / **76.4** |
| Bench | 4.54 / 48.3 | 4.31 / 39.5 | 3.70 / 53.9 | 3.56 / 58.2 | **3.39** / **61.1** | 3.58 / 55.3 |
| Avg. | 5.58 / 59.4 | 5.86 / 45.2 | 4.27 / 65.4 | 4.10 / 67.5 | 4.08 / 68.3 | **3.90** / **70.6** |

Table 9: Quantitative results of noisy shape completion on ScanNet objects of novel categories.

| CD↓ / IoU↑ | 3D-EPN | Auto-SDF | PatchComplete | DiffComplete | SC-Diff | Ours |
|---|---|---|---|---|---|---|
| Bag | 8.83 / 53.7 | 9.30 / 48.7 | 8.23 / 58.3 | **7.05** / 48.5 | 7.41 / 50.0 | 7.56 / **63.7** |
| Lamp | 14.3 / 20.7 | 11.2 / 24.4 | 9.42 / 28.4 | 6.84 / 30.5 | **6.39** / 33.2 | 7.92 / **38.8** |
| Bathtub | 7.56 / 41.0 | 7.84 / 36.6 | 6.77 / 48.0 | 8.22 / **48.5** | 8.09 / 48.4 | **6.42** / 46.9 |
| Bed | 7.76 / 47.8 | 7.91 / 38.0 | 7.24 / 48.4 | 7.20 / 46.6 | 6.91 / 48.6 | **6.42** / **51.1** |
| Basket | 7.74 / 36.5 | 7.54 / 36.1 | 6.60 / 45.5 | 7.42 / 59.2 | 6.38 / **62.2** | **5.62** / 48.2 |
| Printer | 8.36 / 63.0 | 9.66 / 49.9 | 6.84 / 70.5 | **6.36** / **74.5** | 7.10 / 69.1 | 6.59 / 71.1 |
| Avg. | 9.09 / 44.0 | 8.90 / 38.9 | 7.52 / 49.5 | 7.18 / 51.3 | 7.04 / 51.9 | **6.75** / **53.3** |

Objaverse. During fine-tuning, we update the entire model, and we also experimented with LoRA-based fine-tuning (Hu et al., 2021), which showed no performance improvement.

**Results postprocessing.** Since the ground truth shape in the PatchComplete benchmark is obtained from low-resolution TSDF fusion, directly downsampling our model's high-resolution predictions to a low resolution would result in geometries with characteristics different from those produced by TSDF fusion, such as differing geometric details due to low-resolution artifacts. Therefore, we adopt a downsampling method consistent with the construction of the GTs. We first extract the surface meshes from the restoration results and then perform virtual rendering and TSDF fusion to obtain a low-resolution TSDF grid, which is then used as the input for evaluation in the PatchComplete benchmark. When evaluating our model on the ScanNet subset of the PatchComplete benchmark, following the evaluation strategy of AutoSDF (Mittal et al., 2022) in PatchComplete, we take 5 samples for each restoration and keep the best-performing one for evaluation, which is similar to the computation of Minimum Matching Distance (MMD). This is because the ScanNet subset is highly ambiguous; only taking 1 sample leads to a high variance in the evaluation results.

**Full results.** Due to space limitations in the main text, we only presented the average metrics of our method across all categories in the ShapeNet and ScanNet subsets. We provide the complete quantitative results in Tab. 8 and Tab. 9. The proposed method achieves the best results on both CD and IoU on most of the novel categories. It is important to note that the ground truth geometry in the PatchComplete benchmark has a relatively low resolution, which may hinder our method from effectively showcasing its advantages in categories with thin structures. This could also introduce biases, particularly in categories like Lamp and Bench.

