# OpenReview forum: "UniRestore3D: A Scalable Framework For General Shape Restoration"
_ICLR.cc/2025/Conference — ICLR 2025 Poster_

### Official Review · Reviewer_64j6 · 2024-11-01

**Soundness:** 3
**Presentation:** 3
**Contribution:** 3
**Rating:** 8
**Confidence:** 4

**Summary:**

This paper proposed a new framework for general shape restoration, which aims to handle multiple types of defection in a single framework. The proposed framework supports four types of shape defection, namely noise / noise-free completion, noise refinement (i.e., denoising), and super-resolution. The proposed new framework including a new large-scale dataset with various types of defections, a multi-scale defection-robust shape encoder, and a conditional latent diffusion model for shape generation. Experiments on multiple shape restoration tasks demonstrated the effectiveness of the proposed framework.

**Strengths:**

- The attempt of unified framework for 3D shape restoration under different defections is interesting.

- The proposed large-scale shape restoration dataset is very useful to the community.

- Outstanding results on multiple shape restoration tasks which outperforms state-of-the-art.

- Good writing and enough details in the appendix for reproducible research.

**Weaknesses:**

- While I really appreciate the efforts for building such a large and general framework (and the result is really promising), I found it hard to understand the key challenge and insights for building such a framework. Specifically,

(1) What is the key challenge for building the large scale dataset for shape restoration? The process mainly consists of randomly adding different type of pre-defined perturbation on existing large-scale 3D object datasets. It is surprising (or maybe I missed some work?) that no one ever did this before.

(2) As mentioned in the introduction section, "These diverse tasks require different model capabilities, making it challenging to design a
unified model for general shape restoration" - how does the proposed framework addressed this capabilities requirement? It reads something like "we just merged all the data with different task together for training, and it worked."  For example, one direct approach of training a unified framework would be just simply merging all existing dataset for different tasks and then perform joint training. I wonder what the result would be, as it would give us more insight regarding why the proposed framework has good performance - is it simply because of the amount of dataset in the proposed framework is larger, or because of the carefully designed method of defection synthesizing pipeline, or other factors?

- There are some unclear part regarding the dataset and the experiment setup. Please see the "Questions" section below.

**Questions:**

(1) For dataset creation, does every intact shape in the database only employs one of the four subtasks, or some shape will be simultaneously corrupted by multiple subtasks together? Also, for the ablation study in table 4, does the "joint training" has the double amount of data compared to "noisy refinement only" and "noisy completion only"? In other words, does the improvement of the proposed method comes from scale-up of data amount or from joint subtasks learning?

(2) I found it hard to understand the importance of the "scalable training strategy" proposed in section 4.1. Why it is "scalable"? It is mentioned in the paper that "it significantly slows down training due to the need to load high-resolution defective shapes". Isn't it true that loading intact shape should also slows down training in the first stage? Also, the proposed method still requires loading defective shapes in the second stage - so what is improved here?

(3) Will the proposed dataset open-sourced in the future?

---

> ### Author Response · Authors · 2024-11-22
> **Reply to reviewer 64j6 (part 1)**
>
> We appreciate Reviewer 64j6 for recognizing the contributions of our paper and offering valuable comments. Our responses to the feedback are provided below.
>
> ---
>
> **W1: Key challenge for building a general shape restoration dataset**
> > "What is the key challenge for building the large scale dataset for shape restoration? The process mainly consists of randomly adding different type of pre-defined perturbation on existing large-scale 3D object datasets. It is surprising (or maybe I missed some work?) that no one ever did this before."
>
> **WA1:** There are indeed existing datasets aimed at addressing specific subtasks, such as noise-free shape completion (AutoSDF, AdaPointTr) and noisy shape completion (3D-EPN, PatchComplete). However, these datasets face a key challenge: their heterogeneity (e.g., using different representations, varying resolutions, different methods for computing the intact representations, as summarized in the table below) prevents them from being combined for joint training and evaluation, unlike image datasets. To address this challenge, we devoted significant effort to constructing a comprehensive dataset specifically for this purpose. This involved unifying representations across different subtasks, designing tailored construction methods to ensure the synthetic data closely resembles real-world scenarios, combining basic subtasks to enhance data diversity and task complexity, and implementing these processes at scale. This extensive effort enabled us to explore the application of shape generative models in general shape restoration more effectively.
>
> | **Dataset / Attributes** | **Subtask** | **Defective shape repr.** | **Defective shape res.** | **Intact shape repr.** | **Intact shape res.** | **Intact shape computation** |
> | --- | --- | --- | --- | --- | --- | --- |
> | **3D-EPN** | Noisy shape completion | TSDF | $$32^3$$ | UDF | $$32^3 / 128^3$$ | Distance field transform [1] |
> | **PatchComplete** | Noisy shape completion | TSDF | $$32^3$$ | TSDF | $$32^3$$ | TSDF Fusion |
> | **AutoSDF / 3DQD** | Noise-free shape completion | SDF | $$64^3$$ | SDF | $$64^3$$ | Xu & Barbič [2] |
> | **AdaPointTr** | Noise-free shape completion | Point Cloud | 2048 points | Point Cloud | 8192 points | Surface sampling |
>
> [1] Amanatides, J., & Woo, A. (1987). A fast voxel traversal algorithm for ray tracing. In Eurographics (Vol. 87, No. 3, pp. 3-10).
>
> [2] Xu, H., & Barbič, J. (2020). Signed distance fields for polygon soup meshes. In Graphics Interface 2014 (pp. 35-41). AK Peters/CRC Press.
>
> ---
>
> **W2: Key challenges for building the framework and insights for why it has good performance.**
> > As mentioned in the introduction section, "These diverse tasks require different model capabilities, making it challenging to design a unified model for general shape restoration" ......
>
> We divide this part into three smaller questions to answer individually, hoping this will address your concerns.
>
> **W2.1: Required model capabilities for general shape restoration and how they are addressed in the proposed framework.**
>
> > As mentioned in the introduction section, "These diverse tasks require different model capabilities, making it challenging to design a unified model for general shape restoration" - how does the proposed framework addressed this capabilities requirement?
>
> **WA2.1:** We think that a model capable of general shape restoration should possess the following capabilities. Below, we outline these capabilities along with our framework's corresponding solutions:
> 1. Semantic understanding of severely damaged and noisy defective shapes (achieving reasonable completion based on the understanding of the defective shapes even without relying on additional text or image inputs);
>       - We train within a conditional generative model framework using a large and diverse set of paired data.
> 2. Robustness to shapes with varying degrees of degradation;
>       - We employ patch-wise encoding in both intact and defective shape encoders.
> 3. Robustness to irregular, high-intensity noise in defective shapes;
>       - We use a unified representation for intact and defective shapes, explicitly aligning the encoding of the defective shape encoder with that of the intact one. This enhances the quality of the shape representation while pre-learning denoising capabilities, improving robustness to noise.
> 4.  Producing shape restoration results that are diverse and realistic while maintaining high fidelity w.r.t. the input shape;
>       -  We propose a scalable training strategy that facilitates accepting high-resolution defective shapes as inputs.
> 5. The ability to generate high-quality restoration results;
>       - We propose a hierarchical shape generative model as the foundational architecture.

---

> > ### Author Response · Authors · 2024-11-22
> > **Reply to reviewer 64j6 (part 2)**
> >
> > **W2.2: Effect of simple joint training, and key challenges for model training.**
> >
> > > What the result would be if we just simply merging all existing dataset for different tasks and then perform joint training.
> >
> > **WA2.2:** As mentioned in WA1, simply merging all existing dataset is infeasible due to data heterogeneity. More importantly, even after constructing such a dataset as we have done, model design and training must address **two key challenges: scalable training and robustness to irregular noise in defective shapes**, which make it infeasible to simply train existing methods on large-scale data. Here, we further explain these two challenges and our proposed solutions to illustrate our contributions.
> >
> > - **Scalable Training:** A critical capability of general shape restoration models is to preserve the undamaged details in the input shape, i.e., high-fidelity w.r.t. defective inputs. This necessitates support for "high-resolution" geometric inputs. In existing methods, "low-resolution" shapes are often used, or "high-resolution" shapes are first downsampled (e.g., to sparse point clouds) before being fed into the model. This preprocessing step inherently introduces shape defects and leads to low-fidelity w.r.t input shapes as shown in Figure 6 & 7. However, using high-resolution shapes directly as input significantly reduces training efficiency, as it requires frequent reading and online encoding of high-resolution data during training (e.g., an uncompressed TSDF grid at a resolution of 256^3 occupies 64 MB memory). This is the primary bottleneck for training efficiency. To address this, we need a method to pre-compress high-resolution defective shapes, allowing the model to process only the compressed encodings during training.
> >
> > - **Robustness to Noise:** Defective shapes often contain highly random noise, making models susceptible to performance degradation when faced with out-of-distribution inputs. To tackle this, we need to design a noise-robust shape encoder that can encode defective shapes with irregular noise patterns, enhancing the model’s robustness.
> >
> > To address these challenges, we designed a noise-robust defective shape encoder and a corresponding training strategy. **The key insight is to construct a unified representation for both defective and intact shapes.** By aligning the encodings of defective shapes (with irregular noise) and their corresponding intact shapes, we achieve two goals: (1) compressing defective shapes and (2) pre-denoising during the compression process. This reduces the difficulty of training the conditional generative model and enhances robustness to noise, as demonstrated in our ablation studies.
> >
> > **W2.3: More insights on why the proposed framework has good performance.**
> >
> > > More insights on why the proposed framework has good performance, is it because more data, defection synthesizing pipeline, or other factors?
> >
> > **WA2.3:** We believe that diverse and multi-task unified training data is the foundation for achieving good performance in general shape restoration, while designing model architectures and training methods that can effectively leverage large-scale data is the key to attaining such performance.
> >
> > In the ablation study, we validated the effectiveness of data diversity and the increase in data volume it brings to the model’s performance, as well as the importance of a noise-robust defective shape encoder for achieving better results.
> >
> > Based on the analysis of the key challenges in achieving general shape restoration from WA1 and WA2.2, increasing the amount of data and diversity is not a trivial task. While we have taken a step toward the goal of general shape restoration, further work is needed to conduct more specific analyses. For example, examining the impact of different subtask data ratios, the specific construction methods, and sources of defective shapes on model performance, among other factors, remains essential.

---

> > > ### Author Response · Authors · 2024-11-22
> > > **Reply to reviewer 64j6 (part 3)**
> > >
> > > **Q1.1: Dataset details**
> > > > For dataset creation, does every intact shape in the database only employs one of the four subtasks, or some shape will be simultaneously corrupted by multiple subtasks together?
> > >
> > > **QA1.1:** Yes, basic shape restoration conditions are combined to form complex defective scenarios. More specifically, factors such as different TSDF grid resolutions, varying degrees of defect, and different levels of noise are randomly combined, resulting in four distinct subtasks. For instance, noisy shape completion, as the most challenging subtask, involves combinations of different resolutions, significant degrees of defect, and substantial noise.
> > >
> > > ---
> > >
> > > **Q1.2: Ablation study details**
> > > > Also, for the ablation study in table 4, does the "joint training" has the double amount of data compared to "noisy refinement only" and "noisy completion only"? In other words, does the improvement of the proposed method comes from scale-up of data amount or from joint subtasks learning?
> > >
> > > **QA1.2:** The current ablation study does not include a comparison where the training data for a single subtask is equally augmented. Since one of our insights is that unifying different subtasks can lead to more available data, and further augmenting the training data for a single subtask and conducting training would be time-consuming, we plan to explore this comparison in the future. This will allow us to further analyze the impact of multi-task learning and data amount on the final performance.
> > >
> > > ---
> > >
> > > **Q2: Importance of scalable training strategy**
> > > > I found it hard to understand the importance of the "scalable training strategy" proposed in section 4.1. Why it is "scalable"? It is mentioned in the paper that "it significantly slows down training due to the need to load high-resolution defective shapes". Isn't it true that loading intact shape should also slows down training in the first stage? Also, the proposed method still requires loading defective shapes in the second stage - so what is improved here?
> > >
> > > **QA2:** The scalable training strategy refers to the efficient training of the conditional generative model, specifically the hierarchical latent diffusion model (H-LDM). Our method involves three main training stages:
> > > 	(1)	Training the base shape VAE on intact shapes.
> > > 	(2)	Fine-tuning the base shape VAE on defective shapes to obtain a noise-robust defective shape encoder.
> > > 	(3)	Training the H-LDM as the conditional generative model.
> > >
> > > Among these, training the H-LDM is the most time-consuming step. As discussed in WA2 and shown in the table below, reading high-resolution defective shapes online during this stage is impractical. To address this, after completing stages (1) and (2), we compress both intact and defective shapes into latent representations. This significantly reduces both the IO overhead and the computational cost of encoding high-resolution shapes during H-LDM training, making it feasible to train on large-scale datasets.
> > >
> > > |       Model        |  w/ scalable |             | w/o scalable |             |
> > > |:------------------:|:------------:|:-----------:|:------------:|:-----------:|
> > > |                    |    speed     |    time     |     speed    |     time    |
> > > |  H-LDM (coarse)    |  4.37 it/s   |  3.5 days   |   0.22 it/s  |   70 days   |
> > >
> > > Additionally, stages (1) and (2) are simpler tasks that focus more on compression than generation, making them easier to generalize. As the dataset size grows, there is no need to always retrain or fine-tune the base shape VAE or noise-robust encoder. We only need to continue training the H-LDM on pre-compressed encodings, thus enhancing the scalability of our approach.
> > >
> > > In summary, the scalable training strategy significantly improves the efficiency of H-LDM training, making it more feasible to train on large-scale datasets.
> > >
> > > ---
> > >
> > > **Q3: Open-source**
> > > > Will the proposed dataset open-sourced in the future?
> > > - Yes, we plan to open-source the dataset creation pipeline.

---

> > > > ### Comment · Reviewer_64j6 · 2024-11-25
> > > >
> > > > Thanks for the detailed response. I believe my issues are mostly addressed and I have raised my score.

---

> > > > > ### Author Response · Authors · 2024-11-25
> > > > >
> > > > > Thank you once again for your insightful review and constructive feedback on our work. We sincerely appreciate your thoughtful suggestions, which have been instrumental in improving the quality of our paper. These include:
> > > > >
> > > > > 1. Regarding the questions about the key challenges and insights of the framework and dataset: This prompted us to revise the introduction to provide a clearer explanation of these challenges and insights.
> > > > > 2. We recognize that the initial draft did not provide sufficient background on the scalable training strategy. In response, we have revised the relevant paragraph in the method to enhance clarity.
> > > > > 3. To address your question about the combinations of subtasks in the dataset, we have updated the dataset section with a explanation of the possible subtask combinations.
> > > > > 4. As you suggested, we plan to conduct more in-depth studies in the future to explore how variations in subtask data volumes and specific subtask construction methods affect the model's performance.

---

### Official Review · Reviewer_XP48 · 2024-11-02

**Soundness:** 3
**Presentation:** 3
**Contribution:** 4
**Rating:** 8
**Confidence:** 3

**Summary:**

The paper presents a dataset/benchmark that unifies shape completion, denoising and superresolution on TSDFs and a hierarchical latent diffusion based method to solve all those tasks jointly.
The dataset is assembled from a variety of dataset, including Objaverse using a unique pipeline to get from non-manifold-meshes to ground-truth and different versions of incomplete or corrupted TSDF grids.
Their unified restoration method uses two main modules: A hierarchical (multi level) Variational Autoencoder and a hierarchical (multi level) latent diffusion model. All models (except for the coarsest level) are implemented using sparse convolutions, which only act on SDF-values close to the surface. The VAE is trained separately from the diffusion model in two stages. First on the clean shapes. In a second step the VAE is refined on the corrupted shapes, where the latents are guided to be close to the ground-truth latents of the clean objects.
The latent diffusion model runs from coarse to fine. It is conditioned on the latents of the corrupted shape and the occupancy grid predicted in the next coarser level. The diffusion model is trained to denoise the latents into the clean grond truth latents. The occupancy grid for the next level is extracted by decoding the latents into a TSDF using the VAE-decoder and extracting the geometry. On the finest level the decoded TSDF is the final result of the method.
The method is validated on a test-split of the own dataset and also on the 3DQD and PatchComplete benchmarks and demonstrate sota performance.

**Strengths:**

* Sane method with reasonable justification for each step.
* The paper presents a new benchmark for shape completion using TSDFs, which could potentially be very useful in the field, as currently used benchmarks are either not tailored to TSDFS or extremely low resolution.
* The detailed account for all the work necessary to compare to the existing 'benchmarks' beautifully highlight the hideous sate of the benchmarks in this field. Every single paper that is published uses their own internal representation and rely on a very specific data preprocessing. They all have to compare to absolutely terrible benchmarks like Patch Complete or 3DQD - burning so much time and brain of talented people just to figure out how to correctly recreate the benchmark to work with their internal data and somehow stay comparable to the 32^3 numbers. It is an absolute shame. I think the paper did this well, still I wonder what brilliant work the authors could have done with the time they had to waste on this.
* Diffusion based method can produce multiple suggestions which the user can chose from.

**Weaknesses:**

* The shape representation (TSDF) is mentioned for the first time on page 4. Your readers might have very different backgrounds and very different understanding of 3D-Shapes. Some might think you work on 3D-Pointclouds, others think you work with meshes, yet others might think you for sure use binary occupancy grids or unsigned distance fields neural representations or whatever. In the related works you put yourself next to all kind of methods without pointing out this major difference. In my humble opinion the choice of shape representation is a central feature, which defines it's usefulness for different problems. It should be communicated very clearly - if not in the Title at least in the Abstract or at least in the Introduction. Also your resolution of 256 is really nice and also could be mentioned earlier.
* No information about training time of the modules is given. No information of inference time and memory requirement for inference are given.
* Diffusion based method is not usable for online real-time applications (during 3D-scanning).

**Questions:**

* Line 429: typo: tsdf -> TSDF
* Please provide training hardware, memory and time requirements. Same for inference.
* Paper claims that the denoiser can be conditioned on "text or images", which was never demonstrated. Claim must be removed or proven in the given context. If it is not used it should be removed from the method (especially equation 1). If it is used its effect must be evaluated. This is my biggest concern currently.

---

> ### Author Response · Authors · 2024-11-22
>
> We appreciate Reviewer XP48 for recognizing the contributions of our paper and offering valuable comments. We are especially grateful to Reviewer XP48 for the careful review of the evaluation details across different benchmarks, highlighting the challenges caused by the lack of standardization in existing benchmarks. We also appreciate the recognition of the importance of our proposed benchmark and the effort we put into evaluating our method on relevant benchmarks. Our responses to the feedback are provided below.
>
> **W1: Clearly state the shape representation and its resolution early in the writing.**
> > The shape representation (TSDF) is mentioned for the first time on page 4. Your readers might have very different backgrounds and very different understanding of 3D-Shapes. Some might think you work on 3D-Pointclouds, others think you work with meshes, yet others might think you for sure use binary occupancy grids or unsigned distance fields neural representations or whatever. In the related works you put yourself next to all kind of methods without pointing out this major difference. In my humble opinion the choice of shape representation is a central feature, which defines it's usefulness for different problems. It should be communicated very clearly - if not in the Title at least in the Abstract or at least in the Introduction. Also your resolution of 256 is really nice and also could be mentioned earlier.
>
> **WA1:** Thank you for your suggestion! We have emphasized the shape representation in the revised abstract. We also emphasize the benefit of accepting high resolution TSDF grids as inputs in the revised related work.
>
>
> ---
> **W2: More details on training & inference**
> > No information about training time of the modules is given. No information of inference time and memory requirement for inference are given. (and in Questions: Please provide training hardware, memory and time requirements. Same for inference.)
>
> **WA2:** Thank you for pointing out this omission. We have provided relevant information in the A.3 section of the revised appendix.
>
> ---
> **W3: Feasibility on real-time applications**
> > Diffusion based method is not usable for online real-time applications (during 3D-scanning).
>
> **WA3:** Currently, due to speed limitations, our method is indeed not suitable for real-time applications. However, our pipeline is compatible with diffusion acceleration techniques (e.g., better samplers, diffusion model distillation) and other modeling optimizations (e.g., flow matching). Combining these approaches in the future could enable online shape restoration of 3D reconstruction or provide shape priors for real-time 3D reconstruction.
>
> ---
>
> **Q1: Concern about claim of being able to condition on "text or images"**
> > Paper claims that the denoiser can be conditioned on "text or images", which was never demonstrated. Claim must be removed or proven in the given context. If it is not used it should be removed from the method (especially equation 1). If it is used its effect must be evaluated. This is my biggest concern currently.
>
> **QA1:** We indeed used text condition on our evaluation on the 3DQD benchmark. Since the 3DQD benchmark evaluates category-conditional models, not using category condition could lead to ambiguities in the restoration results for categories with similar structures (e.g., tables, chairs, sofas). By incorporating category condition through text condition, our model avoids such ambiguities.
>
> We did not use image condition in any experiments, as we believe it becomes difficult to determine whether the model's restoration relies primarily on the image information or its understanding of the defective shape. Accordingly, we have removed the phrase 'can be conditioned on images' from the paper and retained only 'text.'

---

> > ### Comment · Reviewer_XP48 · 2024-11-29
> >
> > Unfortunately I was not able to participate actively in the discussion unforeseen private reasons.
> > However I think the changes you made based on the feedback from the reviewers have clearly improved the paper. You addressed all my concerns (though in line 348 you missed mention of the image conditioning) and the relevant questions from other reviewers in your rebuttals. I think your approach is solid and better than what is currently published and your new benchmark will be invaluable for future research. I therefore can now wholeheartedly embrace your work and will update my score accordingly.

---

> > > ### Author Response · Authors · 2024-11-29
> > >
> > > Thank you once again for taking the time to provide a thorough and thoughtful review, as well as for offering constructive feedback that has helped us improve the rigor of our work. We deeply appreciate your careful attention to the details of our paper and your recognition of our efforts. We will address the issue at Line-348 in the subsequent revision.

---

### Official Review · Reviewer_mwfZ · 2024-11-02

**Soundness:** 3
**Presentation:** 3
**Contribution:** 2
**Rating:** 6
**Confidence:** 4

**Summary:**

A unified shape restoration model is proposed for handling multiples types of defects (e.g., incompleteness, noise, sparsity) present in scans of 3D objects. The unified shape restoration model is composed of a patch-wise encoder for locally encoding defective shapes, improving generalization capabilities, and a hierarchical latent diffusion model used for generating intact shapes. To enable robustness to various defect types and improve generalization to novel objects, the proposed model is trained on a newly constructed large scale dataset which contains a variety of shape restoration subtasks. Furthermore, a two-staged training scheme is proposed for accelerating training on high-resolution shapes.

**Strengths:**

- The proposed dataset is an improvement over many of the existing datasets used for evaluating shape completion. Many previous datasets are constructed from the few largest categories from the ShapeNet or PartNet dataset and typically only contain a single type of defect (i.e., incompleteness).  On the other hand, the proposed dataset is much larger scale and contains greater diversity as it is constructed from a diverse set of shape datasets, while also modeling more realistic shape defects present in object scans (e.g., noise + incompleteness).

- The qualitative results seem to suggest the proposed approached does a better job than previous approaches at respecting the fine geometric details present in the partial scans while producing higher quality completions of the missing regions.

**Weaknesses:**

- While a high level description of the H-VAE encoder, H-VAE decoder, and H-LDM are provided, there is no description of the actual architectures for these modules. I would expect to see a more detailed description of what the architectures are at the very least in the appendix.

- A scalable training strategy is posed as a contribution; however, there does not exist any evidence that the proposed strategy is more efficient. A table containing training times between the two strategies would demonstrate the benefits more clearly. If the model truly can’t be trained using the normal/standard strategy, the authors could always train on a subset of the data and report training times for that subset or extrapolate what the training time would be for the entire dataset.

- The model does not seem to generalize all that well or be robust to unknown categories. In Table 1, the model obtains a 2-5x worse MMD/AMD on ABO across the different tasks and about a 100x worse MMD/AMD on GSO. This large drop in performance is observed even for the noisy refinement task which should be an easier task than noisy completion.

- The quantitative results don’t really demonstrate that the proposed model is better. In Table 2, the proposed approach is the only model pre-trained on the large scale dataset, which isn’t really a fair comparison for evaluating model performance. Even when pre-trained on their dataset the model performs similarly or worse to NeuSDFusion, and to actually outperform NeuSDFusion they had to artificially add more task specific examples (noise-free completions) to the pre-training data. It’s not clear whether all baseline methods would improve and outperform the proposed method from a similar pre-training. Instead the authors should add a comparison of their approach with no pre-training to Table 2 to fairly evaluate model performance.

- Similarly in Table 3, a comparison needs to be added with no pre-training. The model slightly outperforms SC-Diff, but this improvement could just be from pre-training on a large scale dataset.

**Questions:**

Questions:
- Given that the patch-wise encoders produce sparse feature grids, how is the feature alignment being computed? A shape that has missing geometry will not have the same sparse voxel grid as the intact shape, so is the feature alignment loss only being computed for voxel indices which exist in both sparse feature grids?

- Since all the encoders, decoders, and LDM use sparse convolutions, how is the model able to generate missing structures from the sparse feature grid? Is some form of structure prediction module, similar to Ren et al. (2024) and Huang et al. (2023), being used in the decoder?


Corrections to make:
- In Table 6 of appendix, the best CD for the Lamp category and the best IoUs for the Bed and Bench category are incorrectly bolded.
- In Table 7 of appendix, the best IoUs for the Bathtub, Basket, and Printer categories are incorrectly bolded.

References:
- Jiahui Huang, Zan Gojcic, Matan Atzmon, Or Litany, Sanja Fidler, and Francis Williams. Neural kernel surface reconstruction. In Proceedings of the IEEE/CVF Conference on Computer Vision and Pattern Recognition, pp. 4369–4379, 2023.
- Xuanchi Ren, Jiahui Huang, Xiaohui Zeng, Ken Museth, Sanja Fidler, and Francis Williams. Xcube: Large-scale 3d generative modeling using sparse voxel hierarchies. In Proceedings of the IEEE/CVF Conference on Computer Vision and Pattern Recognition, pp. 4209–4219, 2024.

---

> ### Author Response · Authors · 2024-11-22
> **Reply to reviewer mwfZ (part 1)**
>
> We appreciate Reviewer mwfZ for recognizing the contributions of our paper and offering valuable comments. Our responses to the feedback are provided below.
>
> ---
> **W1: lack of details of network architectures**
> > While a high level description of the H-VAE encoder, H-VAE decoder, and H-LDM are provided, there is no description of the actual architectures for these modules ......
>
> **WA1:** We have added a section of the model architecture in the updated appendix, we hope this resolves your concerns.
>
> ---
> **W2: training time details**
> > A scalable training strategy is posed as a contribution; however, there does not exist any evidence that the proposed strategy is more efficient ......
>
> **WA2:** Thank you for your suggestion. We can more intuitively demonstrate the importance of the proposed scalable training strategy by comparing the training speeds under two different settings. As shown in the table below, taking single-stage H-LDM as an example, if we do not pre-train an encoder to compress the defective shapes to be restored during training, the training speed slows down by nearly 20 times. This results in unacceptable training times and makes it difficult to train with larger-scale data.
>
> | Model | w/ scalable | | w/o scalable | |
> |:--:|:--:|:--:|:--:|:--:|
> |  |  speed | time | speed | time |
> |  H-LDM (coarse) | 4.37 it/s | 3.5 days | 0.22 it/s | 70 days |
>
> ---
>
> **W3: Generalization or robustness to unknown categories**
> > In Table 1, the model obtains a 2-5x worse MMD/AMD on ABO across the different tasks and about a 100x worse MMD/AMD on GSO. This large drop in performance is observed even for the noisy refinement task which should be an easier task than noisy completion."
>
> **WA3:** Achieving robust restoration on unknown categories with severe incompleteness is highly challenging. In our benchmark, we do not utilize text or image inputs; the model must implicitly recognize the semantics of defective shapes and perform restoration based on this understanding. This is inherently infeasible for unknown categories with large missing regions and not the focus of our work. We expect improvement on unknown categories by enlarging the dataset to cover more categories or by fine-tuning text-to-3D models pre-trained on larger-scale datasets. In contrast, one of our goals is to validate the effectiveness of unifying data sources of multiple subtasks compared to single-task data. This has been demonstrated for both known categories and in-the-wild categories in our ablation studies.
>
> However, after careful examination, we discovered that during evaluation on the GSO data, there was an issue with the metrics due to differences in object pose orientation between our predictions and the ground truth. We have corrected the GSO metrics in Table 1. Thank you very much for catching this and we sincerely apologize for not noticing this earlier. With the corrected metrics, the GSO scores are better, and the performance drop in noisy refinement and shape super-resolution is indeed significantly less than in the more challenging noise-free and noisy shape completion subtasks.
>
> Furthermore, the shape refinement task is not always easier. During data construction, we randomly combined different basic shape restoration types. The noisy refinement subtask is randomly paired with different resolutions, and in cases of low resolution (e.g., $32^3$), severe impairment may occur at lower resolutions, making it highly challenging to restore. As shown in the table below, on the shape refinement subtask of the GSO data, our method performs worse on low-resolution defective shapes than on high-resolution ones.
>
> | Resolution | AMD ↓ | MMD ↓ | TMD ↑ |
> |--|--|--|--|
> | 32| 0.518 | 0.347 | 0.489 |
> | 64 | 0.457 | 0.357 | 0.358 |
> | 128 | 0.463 | 0.353 | 0.373 |
> | 256 | 0.211 | 0.161 | 0.203 |

---

> > ### Author Response · Authors · 2024-11-22
> > **Reply to reviewer mwfZ (part 2)**
> >
> > **W4: Fairness of experiments on 3DQD and PatchComplete**
> > >The quantitative results don’t really demonstrate that the proposed model is better. In Table 2, the proposed approach is the only model pre-trained on the large scale dataset, which isn’t really a fair comparison for evaluating model performance. ...... It’s not clear whether all baseline methods would improve and outperform the proposed method from a similar pre-training ......"
> >
> > **WA4:** Our primary goal is not to design a model that outperforms baselines on small-scale datasets but to develop a pipeline that facilitates training with large-scale data. Training or pre-training on large-scale, high-resolution datasets is one of our core contributions. Baseline methods cannot pre-train on large-scale data as we have done. To pre-train on high-resolution defective shapes like ours, additional model and training designs are needed --- more specifically, our proposed scalable training strategy and noise-robust encoder. Without these, the training speed would be too slow, making the entire pipeline impractical. Let us restate the experimental background on 3DQD and PatchComplete to clarify our experimental goals and rationale.
> >
> > - **Our goal:** We aim to process high-resolution defective shape inputs and generate high-resolution intact shapes as outputs. Handling high-resolution defective shapes, rather than downsampling them to low-resolution representations like existing methods do, is crucial. Using the original, high-resolution geometry is necessary to achieve high fidelity relative to the defective shape.
> > - **Existing benchmark setup:** The 3DQD and PatchComplete benchmarks and related baselines use low-resolution defective shape inputs and produce low-resolution intact shapes.
> > - **Experimental background:** We train on high-resolution data and fine-tune and test on the low-resolution data of existing benchmarks to verify the effectiveness of our pre-training on large-scale, multi-task unified data. We are not primarily aiming for exceptional results in 'generating low-resolution intact shapes'; instead, we are addressing a more challenging problem. The existing benchmarks cannot effectively assess our method’s fidelity relative to high-resolution defective shapes or the quality of the high-resolution results generated. Relevant outcomes are presented in Figures 6 and 7.
> >
> > In summary, achieving superior performance without pre-training is not the focus of our work, and pre-training in our setup is infeasible for baselines. Our experiments on 3DQD and PatchComplete demonstrates that pre-training on diverse datasets facilitated by the proposed pipeline enables fine-tuning to surpass state-of-the-art methods on specific subtask datasets.
> >
> > ---
> >
> > **Q1: Details about feature alignment**
> > > Given that the patch-wise encoders produce sparse feature grids, how is the feature alignment being computed? A shape that has missing geometry will not have the same sparse voxel grid as the intact shape, so is the feature alignment loss only being computed for voxel indices which exist in both sparse feature grids?"
> >
> > **QA1:** We compute the feature alignment loss only on the features corresponding to sparse voxels shared by both defective and intact shapes. We believe this approach provides sufficiently dense supervision for effective training.
> >
> > ---
> > **Q2: Generation of missing structures**
> > > Since all the encoders, decoders, and LDM use sparse convolutions, how is the model able to generate missing structures from the sparse feature grid? Is some form of structure prediction module, similar to Ren et al. (2024) and Huang et al. (2023), being used in the decoder?"
> >
> > **QA2:** Yes, we use a similar strategy by subdividing the voxel grids and adaptively pruning some structure.

---

> ### Comment · Reviewer_mwfZ · 2024-11-25
>
> Thank you for providing such in depth responses. I have some additional comments below pertaining to the answers that have been provided.
>
> ---
> **W2:  training time details**
> Please include this table either in the ablations section of your main paper or in the appendix. This table is the only result that provides clear evidence for your claimed contribution of "scalable training", so this is an important result to include in the paper.
>
> ---
> **W3: Generalization or robustness to unknown categories**
> Thank you for providing the updated numbers on the GSO dataset. The new results look significantly better and address my concerns in regard to generalization. Additionally, it was an oversight on my part that the noisy refinement task also contained varying resolution inputs. I originally assumed they were all at the 256^3 resolution, but the varying input resolution does help explain the potential difficulty in the refinement task. I suggest including the performance break down table of resolution for the noisy refinement task in the appendix, as I believe this is an interesting experiment on how performance varies with varying levels and types of shape degradation.
>
> ---
> **W4: Fairness of experiments on 3DQD and PatchComplete**
> Thank you for clarification on the claims for why your model is better over existing architectures. I acknowledge that designing a model architecture which enables support for training on higher resolution data and significantly larger datasets are certainly improvements over existing architectures which cannot support this.
>
> ---
> **Q1: Details about feature alignment**
> Please add this detail to the paper. It is not clear that this is how the feature alignment loss is being computed from the loss function alone.
>
> ---
> Overall, I have decided to raise my score as I believe the contributions presented (e.g., new dataset, model which can generate high resolution completions, and support for training on large scale datasets) are clear benefits over the existing work on shape restoration. Please do still update the final version of the paper to include the suggestions I have mentioned above though.

---

> > ### Author Response · Authors · 2024-11-25
> >
> > Thank you once again for taking the time to provide a thorough and thoughtful review, as well as for your insightful and constructive feedback. We are especially grateful for pointing out the issue regarding the GSO data in Table 1. Your review has prompted us to conduct a deeper analysis of our method. We will incorporate the three revision suggestions you provided into our paper and will notify you once the revisions are complete.

---

### Official Review · Reviewer_Bu6F · 2024-11-04

**Soundness:** 3
**Presentation:** 3
**Contribution:** 3
**Rating:** 5
**Confidence:** 3

**Summary:**

This paper presents a unified model for general shape restoration, aiming to recover 3D shapes with various defects, such as incompleteness and noise. By standardizing data representation and constructing a large-scale dataset of diverse shape defects, the authors develop an efficient hierarchical generation model and a noise-robust encoder, demonstrating improved applicability and scalability across multiple restoration subtasks on various datasets.

**Strengths:**

The problem is useful, for preprocessing noisy 3D scans.
The results seem good.

**Weaknesses:**

There are no visual comparisons for other methods, making it hard to compare the improvements.
All data is from the proposed dataset, how about results on real noisy 3D scans?
How about the results on other datasets? Does the network overfits on training set?
There is no overview figure of the whole method, from Figure 3-4, it is still hard to follow the method design.
For different noises, like low-resolution, noisy completion, noisy refinement etc., do they share the same pipeline/network?

**Questions:**

See above.

---

> ### Author Response · Authors · 2024-11-22
>
> We appreciate Reviewer Bu6F for recognizing the contributions of our paper and offering valuable comments. Our responses to the feedback are provided below.
>
> ---
>
> **W1: Experimental results across different datasets**
> > W1.1: "There are no visual comparisons for other methods, making it hard to compare the improvements."
>
> > W1.2: "All data is from the proposed dataset, how about results on real noisy 3D scans?"
>
> > W1.3: "How about the results on other datasets?";
>
> > W1.4: "Does the network overfits on training set?"
>
> **WA1:** We conducted experiments on three distinct benchmarks, each involving datasets with sources different from our training dataset and covering various experimental settings. We regret any misunderstanding Reviewer Bu6F might have had regarding the experiments in the paper, and we provide further clarification below. As illustrated in Figure 2, we categorize shape restoration into four basic types: Noise-free Completion, Super-resolution, Noisy Refinement, and Noisy Completion.
> - Sec 5.1, Table 1, Figure 5: These present the results tested on our newly proposed dataset/benchmark. This benchmark is designed to evaluate performance in the new general shape restoration setting, which includes all four basic types mentioned above. It requires a single model to handle these shape restoration scenarios simultaneously. Moreover, the test data (from GSO and ABO) is distinct from the training data (from Objaverse and ShapeNet).
> - Sec 5.2, Table 2, Figure 6: These show the performance of our method on noise-free completion, a single shape restoration setting tested on an existing benchmark independent of our proposed dataset.
> - Sec 5.3, Table 3, Figure 7: These demonstrate the performance of our method on noisy shape completion, another single shape restoration setting tested on an existing benchmark, also independent of our proposed dataset.
>
> More specific clarifications addressing the four sub-questions:
> - **WA1.1:** In Figure 6 and Figure 7, we provide qualitative comparisons between our method and other baselines, clearly illustrating the strengths of our method.
> - **WA1.2 & WA1.3:** Table 2 and Table 3 present the quantitative results on different benchmarks (i.e., datasets). Notably, Table 3 includes results on real noisy 3D scans from the ScanNet dataset, showcasing the effectiveness of our method in real-world noisy scenarios.
> - **WA1.4**: The quantitative and qualitative results presented in Table 1–3 and Figure 5–7 are all based on validation or test sets. We believe this demonstrates that our model does not exhibit significant overfitting, as the performance holds consistently on unseen data.
>
> ---
>
> **W2: Lack of an overview figure**
> > There is no overview figure of the whole method, from Figure 3-4, it is still hard to follow the method design.
>
> **WA2:** In Figure 3, we illustrate the model’s inference pipeline, which primarily consists of the noise-robust encoder for defective shapes and the conditional generative model for generating intact shapes. The training process for our model comprises three key stages: (1) Training the VAE on intact shapes; (2) Training the noise-robust encoder for defective shapes; (3) Training the conditional generative model H-LDM. Figure 4 provides a detailed depiction of the first two stages of training. The training process for the conditional generative model (H-LDM), while crucial, follows standard methodologies and was therefore omitted from the figure for brevity.
>
> We are open to feedback on specific aspects of Figures 3 and 4 that might fall short in providing a comprehensive overview of the method design. This will help us refine and improve the presentation for greater clarity and completeness.
>
> ---
>
> **W3: About problem setup and model capabilities.**
> > For different noises, like low-resolution, noisy completion, noisy refinement etc., do they share the same pipeline/network?
>
> **WA3:** Yes, the primary motivation of our work is to enable a single pipeline/network to handle the restoration of shape defects caused by various scenarios, achieved through the combination of dataset and model design tailored for this purpose.

---

> > ### Comment · Reviewer_Bu6F · 2024-11-25
> >
> > Thanks for the explanations.
> >
> > I further checked the paper, for the first question (concerning Sec 5.1, Table 1, Figure 5). In the paper, it says "We evaluate the model separately on known categories and in-the-wild instances to quantify its capability... " and "The known categories consist of 13 classes from ShapeNet with a substantial number of training samples (from ShapeNet-13)" and "in-the-wild categories are from GSO and ABO". It seems most test sets are from the same dataset with fine-tuning (ShapeNet). Figure 5 did not point out which data is from ShapeNet and which are from un-seem datasets. It would be helpful to clarify. In Table 1, there are no numbers of other methods (those in Table 2), I am inquisitive about the generalization ability of these methods.
> >
> > For Table 2 and Figure 6, 3DQD is a new unseen dataset/benchmark with known and seen categories (in my understanding). Examples in Figure 6 show that given half of a shape, methods can complete the shape in several ways. For the chair data, I think the results of AutoSDF, 3DQD and the proposed one are all reasonable. Maybe comparing with GT is not the best way to evaluate. I would like to hear from authors' opinions.

---

> > > ### Author Response · Authors · 2024-11-25
> > >
> > > Thank you for your further comments; we are pleased to provide additional explanations to address the issues you have raised.
> > >
> > > **Q1: More details about the test sets and results.**
> > >
> > > In Figure 5, only two examples, the car and the lamp, are from known categories; all other examples belong to "in-the-wild" categories. We will revise Figure 5 to highlight this information more clearly. Specifically, ShapeNet-13 includes the following categories: chair, car, sofa, airplane, lamp, telephone, watercraft, loudspeaker, cabinet, table, display, bench, and rifle, which are considered known categories. In Table 1, we present results on three different datasets: GSO, ABO, and ShapeNet. All of them are conducted using a single model without additional fine-tuning.
> > >
> > > **Q2: Baseline results and generalization ability.**
> > >
> > > Baseline methods are limited to specific subtasks of shape restoration, can only handle certain low-resolution geometric inputs, and are not well-suited for efficient training on our proposed large-scale dataset. In contrast, our method addresses the more complex problem of general shape restoration and can be adapted to various existing benchmarks with minimal fine-tuning. Therefore, to demonstrate its effectiveness, we compare our model with baseline methods on established benchmarks. As shown in the table below, our approach supports defective shape inputs across multiple resolutions and generates high-resolution intact shapes, making it compatible with other benchmarks, whereas the reverse is not true.
> > >
> > > | **Method** | **Subtask** | **Defective shape repr.** | **Defective shape res.** | **Intact shape repr.** | **Intact shape res.** |
> > > | --- | --- | --- | --- | --- | --- |
> > > | **PatchComplete** / **DiffComplete** | Noisy shape completion | TSDF | $$32^3$$ | TSDF | $$32^3$$ |
> > > | **AutoSDF / 3DQD** | Noise-free shape completion | SDF | $$64^3$$ | SDF | $$64^3$$ |
> > > | **AdaPointTr** | Noise-free shape completion | Point Cloud | 2048 points | Point Cloud | 8192 points |
> > > | **Ours** | General shape restoration | TSDF |  $$\le256^3$$ | TSDF | $$256^3$$ |
> > >
> > > Regarding the generalization ability of baseline methods, they cannot generalize across different subtasks. As for their generalization within their supported single subtasks, we hope the experimental results on 3DQD and PatchComplete provide relevant insights. The PatchComplete benchmark mainly focuses on novel-category completion ability. As shown in Tab.3 and Fig. 7, our method achieves better results than PatchComplete on novel categories.
> > >
> > > Overall, for learning-based methods, restoration subtasks with large missing regions (noisy completion, noise-free completion) rely more on the model's extensive semantic understanding and generation capabilities, resulting in poorer generalization on novel categories. In contrast, local restorations (super-resolution, noisy refinement) rely more on local denoising and repair capabilities, leading to better generalization on novel categories. This is also reflected in the results of Table 1, where the metrics for noisy/noise-free completion are generally worse than those for super-resolution/noisy refinement.
> > >
> > > **Q3: About evaluation metrics on 3DQD.**
> > >
> > > Evaluating shape restoration results based on a single ground truth is not perfect, but the current metrics already consider the diversity of generation results. When evaluating on 3DQD, we follow previous works [1,2,3] and assess the MMD, AMD, and TMD metrics. For each shape to be completed, these metrics are computed based on K = 10 samples, focusing respectively on completion quality, fidelity, and diversity. These metrics are correlated, for example, a method needs good diversity (TMD) to increase the probability that one of the K generated results is highly consistent with the ground truth, and thereby leading to a better MMD. Although baseline methods can generate plausible results on the chair category, considering the overall quality across multiple categories, our method indeed achieves higher quality and includes more geometric details. In summary, we acknowledge that the current evaluation metrics are not perfect, but since we remain consistent with previous works, we believe this is reasonable.
> > >
> > > [1] Wu, Rundi, et al. "Multimodal shape completion via conditional generative adversarial networks." ECCV 2020
> > >
> > > [2] Mittal, Paritosh, et al. "Autosdf: Shape priors for 3d completion, reconstruction and generation." CVPR 2022
> > >
> > > [3] Li, Yuhan, et al. "3dqd: Generalized deep 3d shape prior via part-discretized diffusion process." CVPR 2023
> > >
> > > If you have any further questions, please feel free to let us know, and we will do our best to address them.

---

> > > > ### Comment · Reviewer_Bu6F · 2024-11-27
> > > >
> > > > Thanks for the reply. It provides many further details.
> > > >
> > > > In Table 1, the evaluation is done for the proposed method only, without comparisons to other methods (e.g. super-resolution). It is hard to determine from the numbers without comparisons.
> > > >
> > > > After reading the rebuttal and all responses, as well as other reviews, I would maintain my original score.

---

> > > > > ### Author Response · Authors · 2024-11-27
> > > > >
> > > > > Thank you for your reply. As previously mentioned, evaluating separately on the 3DQD and PatchComplete benchmarks is actually more favorable to the baseline methods. If this is insufficient to provide comparative results between the methods, we can test some **pre-trained** baseline methods on our benchmark. Do you think this would be reasonable? We look forward to hearing more details about your expected evaluation setup.

---

> ### Comment · Reviewer_XP48 · 2024-11-29
> **Questioning the Review's Rationale**
>
> I respectfully disagree with the assessment of Reviewer Bu6F. While I understand the importance of rigorous evaluation, I believe the authors have adequately addressed the concerns raised in the original review and subsequent discussions.
>
> Specifically, the authors have provided clear explanations and clarifications to the questions raised by Bu6F. The __new__ benchmark, which is a significant contribution of this paper, is not directly compatible with older works. This becomes clear when considering the efforts described by the authors (appendix) to run their method on the old benchmarks. Including results from prior works in Table 1 would not contribute new insights over what was evaluated on the old benchmarks already.
>
> I encourage Reviewer Bu6F to reconsider their stance, given the authors' efforts to address the raised issues, the novelty of the proposed benchmark and the method which out-scales and outperforms prior work. A more constructive approach would involve providing specific suggestions for improvement, rather than maintaining a negative stance despite the author's efforts.
>
> Ultimately, I believe the paper's contributions warrant a fair and more favorable evaluation.

---

### Comment · Area_Chair_on6G · 2024-11-21
**Please initiate discussions!**

Dear authors and reviewers,

The discussion phase has already started. You are highly encouraged to engage in interactive discussions (instead of a single-sided rebuttal) before November 26. Please exchange your thoughts on the submission and reviews at your earliest convenience.

Thank you,
ICLR 2025 AC

---

### Author Response · Authors · 2024-11-22
**General response to reviewers**

We sincerely thank the reviewers for their time and effort in reviewing our paper and making valuable suggestions. We are pleased that you recognize UniRestore3D addresses a problem that is both useful (Bu6F) and interesting (64j6). It's gratifying to know that  our proposed dataset is considered useful (XP48, 64j6), especially with its larger scale, greater diversity, and realistic shape defects (mwfZ). We also appreciate your positive feedback on our results, which you found to be outstanding (64j6) with better quality and fidelity (mwfZ).

We have uploaded a revised draft of our paper, with changes marked in blue. We summarize the revision here:
- added a new section about network architecture in the appendix;
- added a new section about training & inference details in the appendix;
- clearly stated the capabilities required for a model for general shape restoration in the introduction;
- the penultimate paragraph of the introduction has been rewritten to emphasize the technical challenges of general shape restoration and to highlight our insights;
- the paragraph about the scalable training strategy in Sec. 4.1 has been rewritten to clarify the related challenges and the necessity of such a strategy, making it easier to understand;
- updated results on the GSO dataset in Tab. 1 as we found a mistake in our evaluation code of the GSO dataset;
- streamlined the experimental section by removing some unnecessary details and relocating them to the appendix to meet the page limit；
- There are also several minor modifications based on your valuable suggestions:
    - modify the claim of "being able to condition on text or images" to text only;
    - emphasize the shape representation earlier in the main text;
    - typos and incorrect bold formatting in the appendix tables have been corrected.

We sincerely value your reviews and believe they have greatly contributed to making the new revision more comprehensive. Additionally, we have responded to your individual comments and questions by commenting on your reviews.

---

### Comment · Area_Chair_on6G · 2024-11-25
**Last day for interactive discussions!**

Dear authors and reviewers,

The interactive discussion phase will end in one day (November 26). Please read the authors' responses and the reviewers' feedback carefully and exchange your thoughts at your earliest convenience. This would be your last chance to be able to clarify any potential confusion.

Thank you,
ICLR 2025 AC

---

### Meta-Review · Area_Chair_on6G · 2024-12-20

**Metareview:**

The submission received mostly positive reviews. Although EgpU has some reservations on the visual comparison and validity of Table 1 (without a baseline), the other reviewers generally appreciate the unified framework with strong results, as well as the contribution of a new benchmark dataset for shape restoration. After reading the paper, the reviewers' comments and the authors' rebuttal, the AC agrees with XP48 and and mwfZ that the lack of comparison against other methods in Table 1 are insufficient grounds for rejection, and the merits outweighs the limitations in general. As such, the AC recommends acceptance.

**Additional Comments On Reviewer Discussion:**

The reviewers raised questions regarding issues on comparisons (Bu6F, mwfZ) and clarity in details (Bu6F, mwfZ, XP48, 64j6). The questions were addressed by the authors in good detail. Reviewers mwfZ, XP48, and 64j6 were convinced by the responses and raised their ratings. The AC agrees with the evaluation.

---

### Decision · Program_Chairs · 2025-01-22

Accept (Poster)